# Behaviour and vocalizations of two sperm whales (*Physeter macrocephalus*) entangled in illegal driftnets in the Mediterranean Sea

**Monica Francesca Blasi**[1]*, **Valentina Caserta**[1], **Chiara Bruno**[1], **Perla Salzeri**[1], **Agata Irene Di Paola**[1], **Alessandro Lucchetti**[2]

**1** Filicudi WildLife Conservation, Località Stimpagnato, Filicudi, Lipari (ME), Italy, **2** Centro Nazionale Ricerca - Istituto per le Risorse Biologiche e le Biotecnologie Marine (CNR-IRBBM), Ancona, Italy

☯ These authors contributed equally to this work.

* blasimf@yahoo.com

**Data Availability Statement:** All relevant data are within the manuscript.

**Funding:** The author(s) received no specific funding for this work.

## Abstract

Illegal driftnetting causes each year several entanglements and deaths of sperm whales in different Mediterranean areas, primarily in the Tyrrhenian Sea. In summer 2020, during the June-July fishing season, two sperm whales were found entangled in illegal driftnets in the Aeolian Archipelago waters, Southern Italy. These two rare events were an exceptional chance to collect behavioural and acoustics data about entangled sperm whales. We analysed 1132 one-minute sets of breathing/behavioural data and 1575 minutes of acoustic recording, when the whales were found entangled, during the rescue operation, immediately after release, and in the days thereafter. The first whale was generally quiet showing a general status of debilitation/weakness, numerous skin lesions, and low breathing rate (0.31 (0.60)); it collaborated during rescue operations. On the contrary, the second whale showed a high level of agitation with a high breathing rate (1.48 (1.31)) during both the entanglement period and the net cutting operations, vigorously moving its fluke and pectoral fins, opening its mouth, sideway rolling or side fluking and frequently defecating. Acoustically, the first whale produced mainly single clicks in all phases except for two series of creaks during rescuing operations while the second whale produced a wide range of vocalizations (single clicks, likely either slow clicks or regular clicks, creaks, and codas). Our observations indicate that acoustics, respiratory and behavioural parameters may be useful to monitor the physical/physiological status of sperm whales during disentanglement operations.

## Introduction

The sperm whale (*Physeter macrocephalus*) Mediterranean population, isolated from the Atlantic population as shown by genetic evidence [1, 2], is considered "Endangered" based on the most recent assessments [3, 4]. Indeed, the Mediterranean population is facing a drastic decline over the last half of this century, counting less than 2,500 mature individuals [4–6]. The main threat faced by this species is entanglement by large-scale driftnets fishery [7–9]

**Competing interests:** The authors have declared that no competing interests exist.

classified as "a concerning matter" by the International Whales Commission (IWC) in 1994 [10]. Starting from the 1980s, driftnets have been used on large scale fisheries for decades by numerous fleets of the Mediterranean Sea to catch mainly large pelagic species, such as swordfish (*Xiphias gladius*) and blue tuna (*Thunnus thynnus*) [11–13]. Although driftnets in open waters have been banned by the United Nations since 1992 and the use of these nets of all sizes was prohibited by the European Union from 2002, the illegal use of driftnets continues to be reported in several Mediterranean areas such as Turkey, Algeria, Morocco, Spain, and Italy [6, 7, 9, 12, 14–16]. As a result, a dramatic increase in bycatch mortality of sperm whales in the Mediterranean Sea was observed in the period 1986–2000, from 20–30 estimated cases per year before 1990 to > 100 cases up to 2010 [17]. According to the Italian Database on Cetacean Strandings [18], from 2000 to 2020, 87 sperm whales stranding records have been registered with 105 animals involved. In the last two years, 17 of the 21 sperm whales stranded in Italy were found along the Tyrrhenian Sea coast. This part of the Mediterranean basin is thought to be an important feeding/breeding ground for this species, where both mature/immature males and social units of mature females with their juveniles/calves are regularly encountered [19–22] as a result of the morphological aspects of the bathy-morphological setting, characterized by canyons and seamounts, which have already been positively related to the presence and distribution of the sperm whales [23, 24]. The Aeolian Archipelago located in the Southern Tyrrhenian Sea (Sicily, Italy) is no exception [25]. Several sperm whale encounters (both solitary individuals and social groups) have been documented in this area in the last six years [26]. Moreover, two juvenile male sperm whales were reported to be entangled in illegal driftnets from 2007–2017, and successively stranded dead along the nearest coast as part of the net had not been removed.

Other studies have already described the lethal consequences of entangling large whales in nets along the coast of Ecuador [27, 28], in Southern Brazil [29], and in the Mediterranean Sea [11, 17]. However, behavioural observation and vocalizations of a social unit of sperm whales entangled in illegal driftnets have been previously documented only in the Southern Tyrrhenian Sea [11], but the acoustic analysis was conducted on a social unit as sounds could not be ascribed to individual whales.

In this study, we documented two sperm whales' entanglement events that occurred in the Aeolian Archipelago, during summer 2020. Data on the behaviour, breathing pattern, diving time, and acoustic production of these entangled whales were collected in different phases, from entanglement to post-release.

## Materials and methods

On 26th June 2020 at 10:01 (all the times are in 24h format and related to UTC+1 time zone), a severely debilitated 10 m juvenile male sperm whale (named "first whale") was found entangled by the tail in a driftnet, 6 miles northern of Lipari island (Aeolian Archipelago; Fig 1).

It had several abrasions and scars all over the body, especially in the dorsal area. No signs of starvation were observed.

On 18th July 2020 at 11:43, another juvenile sperm whale of 9 m (named "second whale") was found completely entangled in a driftnet five miles north of Salina (Aeolian Archipelago; Fig 1). Sex was not identified, as the animal was generally very active and nervous not allowing to researchers to retrieve any clear footage of the ventral area. Sign of starvation or severe injuries were not observed.

Behavioural, breathing pattern, and acoustics data were collected throughout the entire duration of the disentanglement process, which has been divided into phases, in order to associate data with the related events. Due to the different approaches adopted by the rescuers and

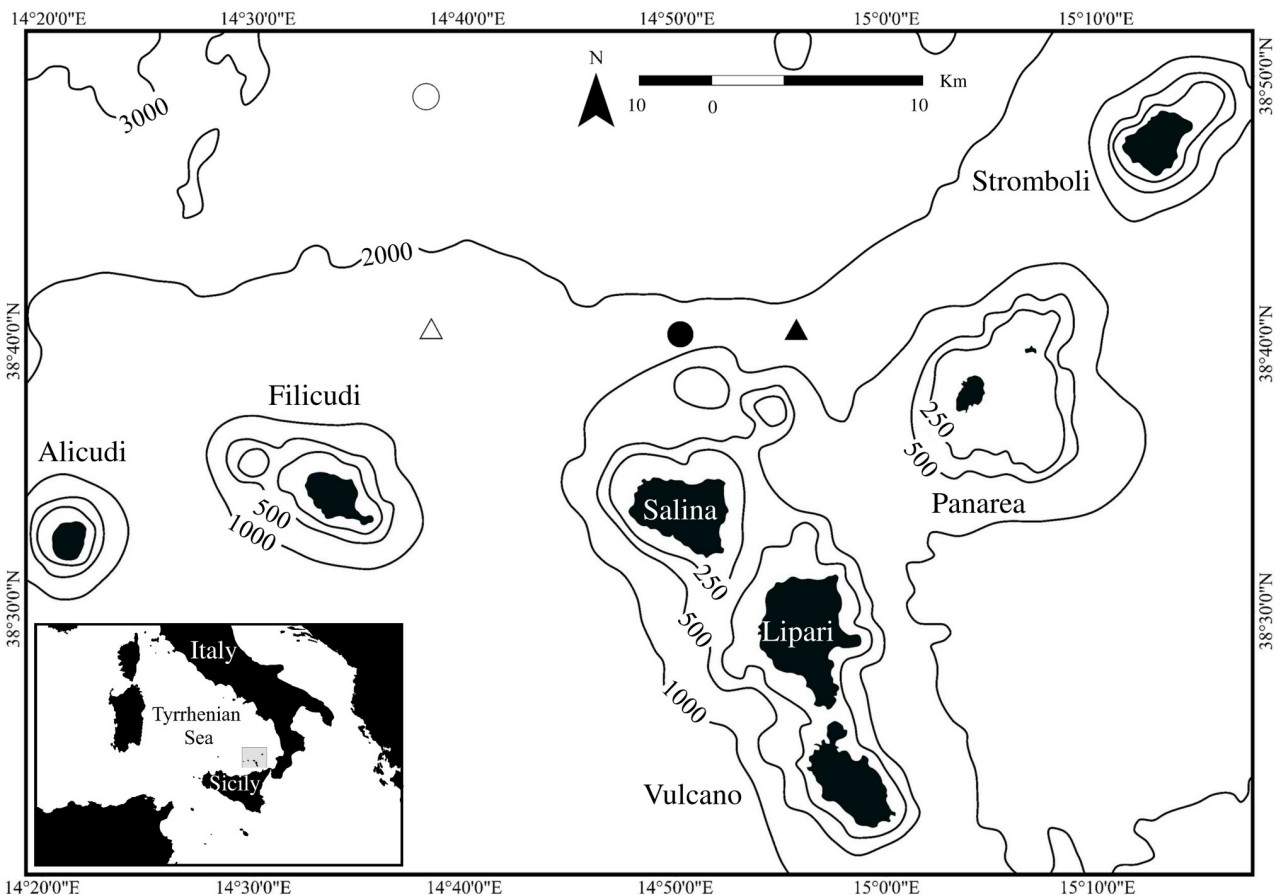

**Fig 1. Map of the Aeolian Archipelago.** Initial (black symbols) and final positions (white symbols) of first (triangle) and second (circle) whales.

the consequent outputs, some phases were in common between the two whales, while others were specific for a single whale. The common phases were described as follow: a) "trapped", including the finding of the entangled animal and waiting for the Italian Coast Guard intervention; b) "rescuing", involving the disentanglement process performed by the scuba rescuers. On the other hand, phases specific to the first whale were: a) "pre-rescue", including the Italian Coast Guard arrival, assessing the situation and pre-disentanglement operations (i.e. immobilizing the tail with a rope and secure it to the main vessel); and b) "post-rescue", involving monitoring after the release (Fig 2).

For the second whale, one specific phase was observed, which was described as "following", in which operators followed with the boat the escaping whale. This phase included a night chasing and no data were collected (Fig 3).

Finally, a "follow-up" monitoring was performed aimed to re-sight the whales.

The behavioural displays of the entangled individuals were recorded using a combination of focal observations (sampling at one-minute intervals) and video recording [30]. Underwater cameras (GoPro Hero7) were used to record the sperm whales' behaviour during the disentanglement process [30–32]. The behaviour of the sperm whales was recorded according to different behavioural activities previously reported in literature [11, 33–35], i.e., open mouthed, side roll, agitation of fluke and pectoral fins, side fluke, lobtail, spy hop, and defecation. The rate of

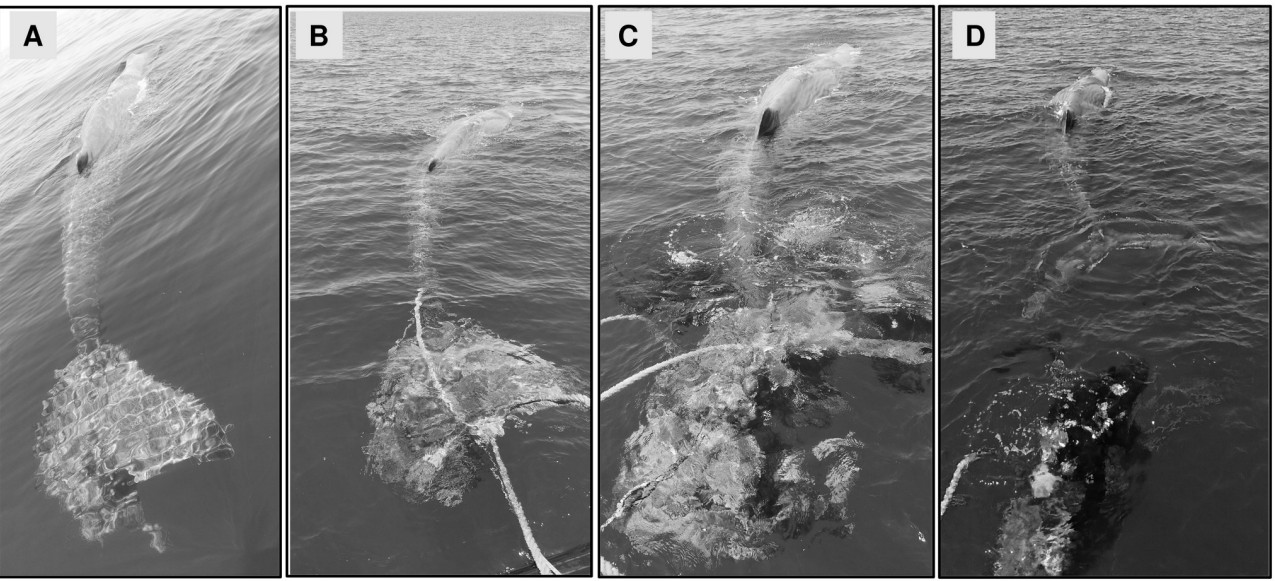

**Fig 2. Disentanglement phases for the first whale.** A) Trapped (only tail). B) Pre-rescue. C) Rescuing. D) Post-rescue.

each behavioural activity (number of records/number of one-minute behavioural sets) was calculated in the different phases for both whales.

The breathing pattern was recorded at one-minute intervals and the mean (SD) breaths per minute (breathing rate) was calculated for each phase. Since the assumption of the homogeneity of variance was rejected, non-parametric Kruskal-Wallis tests and post-hoc testing using un-paired Mann-Whitney U tests with a Bonferroni-adjusted alpha level were used for comparisons of median values in breathing datasets among different phases and between whales. Although both whales displayed surface behaviour, the second whale performed several dives, the dive time of which was recorded [30, 36, 37].

Acoustics offers important insights into the behaviour of sperm whales, providing useful information for rescuing operations [38–42]. The sperm whale vocalizations were recorded using a hydrophone (Aquarian Audio H2a, sensitivity -180 dB re: 1 V/µPa) deployed by the research boat and connected to a wideband solid-state recorder (sampling rate: 48 KHz) at a depth of about 5 m. Differently to behavioural data, acoustics recordings were not constant, because the hydrophone was deployed only in optimal conditions (i.e. minimal background noise due to rescue operations). Emissions were classified into three categories: "single clicks" (SC); "creaks" (CR) and "codas" (CO), according to the click structure described in the literature [43–49]. The acoustical analysis was performed by Raven Lite 2.0.1 software, which has been used to measure upper and lower frequency limits and calculated inter-click intervals (ICI) for each detected sound. In addition, creaks and codas, which are composed by trains of clicks (ICI < 0.3 s), duration of the trains and time interval between trains have been calculated. In this study, "single clicks" were scored for sequences of clicks (ICI > 1 s) that were not organized in a defined structure, as described for "usual clicks" [39]. Kruskal-Wallis tests and Mann-Whitney U tests with post hoc Bonferroni correction were run to single out significant differences in ICIs (for single clicks and creaks) and duration (for creaks only) between whales and among the different phases.

All statistical analyses were performed with R 4.03.

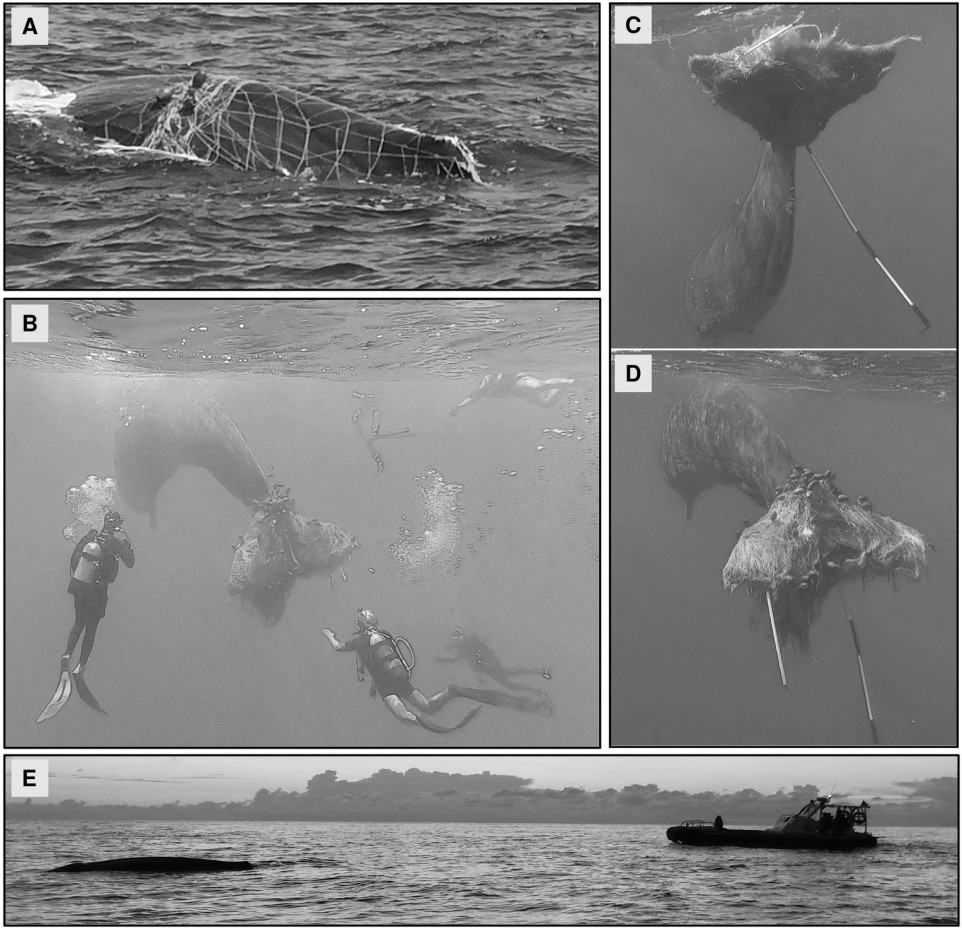

**Fig 3. Disentanglement phases for the second whale.** A) Trapped (all body). B) Rescuing. C) Rescuing (first day) (first attempt of net removal using a boat hook). D) Rescuing (first day) (second attempt of net removal using a boat hook). E) Following.

## Results

### Entanglement events and rescue operations

The first sperm whale was found on the sea surface with its tail entangled and immobilised by the driftnet on 26th June 2020 (Fig 2). For a short period, three other sperm whales were spotted about two miles away from the entangled animal, but no vocalizations were detected. At 12:24, two operators of the Coast Guard started pre-rescue operations to secure the whale's flukes to the main vessel (Fig 2). In particular, the whale was cautiously secured to the main vessel with a rope performing a twisted-pair knot around the caudal peduncle. Rescue operations (disentangling the netting from the tail and cutting the netting twines without hurting the animal, when necessary) started at 14:20 with three divers of a local diving team. The whale remained moderately calm throughout all the rescue procedures and in about 55 minutes it was completely free to move. Subsequently, it stayed close to the divers who frequently touched its body in order to cut the remaining pieces of netting. After release, the animal was closely monitored for two hours, before disappearing (final recorded location is shown in Fig 1). On 29th June 2020, three days after the first sighting, the entire whales' group of four individuals was re-sighted and the whale was observed surfacing and diving regularly (data not shown).

The second sperm whale was found completely entangled and wrapped in the driftnet on 18th July 2020 (Fig 3). No other individuals were sighted or acoustically detected in the proximity of the animal. The rescue operations started at 12:40 and in an hour a Coast Guard diver removed most of the net from the body except for the tail. Then, a second rescue diver attempted to secure the tail to the main vessel, a procedure recommended for small cetaceans by Hamer and Minton [50]. Meanwhile, eight other vessels arrived in close proximity to the animal, but only two of them were rescue boats, which brought seven more divers to support the rescuing operations. The animal got nervous and started to move away from the area breaking the rope connected to the main vessel. Two attempts were made to reach the net using boat hooks, but they got stuck in the tangle of netting. Monitoring activities were performed overnight in order to constantly follow the animal and a lit mark buoy was fixed to the tail. The next morning three more attempts were made by another group of divers, for a total of five boats and ten divers, although the sperm whale started to perform long dives. Therefore, it was difficult to follow its route, so that the whale was last tracked by its blow at 14:38. At 20:00, rescue operations were suspended and scheduled to resume the next day in daylight. In the following days, Coast Guard searched the animal in the area around the last known location, but no results were provided. On 5th October 2020, seventy-nine days after the first sighting, the whale was re-sighted at 17:30, still entangled, at 1 mile from Salina (the same Island where the whale was first spotted) and, as it was very elusive, no rescue operation could be started. Nevertheless, the whale was acoustically monitored for 13 hours during the night.

## Behavioural data

A total of 1,132 one-minute behavioural sets were analysed, 298 for the first whale (26th June 2020) and 834 for the second whale (18th-19th July 2020). The rate of each behavioural activity at each phase was reported for both whales (Fig 4).

The first whale was generally quiet and poorly reactive during entanglement and rescue operations. General status of debilitation and weakness was observed with numerous lesions and wounds on its body and flukes; the whale appeared not to react nervously to the rescue operations, apart from two behavioural categories. i.e., opening of mouth and slight agitation of fluke and pectoral fins (Fig 4A). Defecation was observed only in the post-rescue phase (Fig 4A). On the contrary, the second whale was particularly agitated during both the entanglement period and the rescue operations, vigorously moving its fluke and pectoral fins, frequently opening its mouth, and sideway rolling or side fluking and defecating (Fig 4B). Although the remained attached net, during the following period, the whale was observed lifting the fluke out of the water to bring it down onto the surface of the water in order to make a loud slap (lobtailing). Spy-hopping, breaching, and fluke up were never recorded for both whales after release and throughout the following period.

## Breathing pattern and diving time

The dataset, as for the behaviour, included 1,132 one-minute breaths sets, of which 298 for the first whale and 834 for the second whale. The breathing rate (number of breaths per minute bin) was calculated over different phases showing a different trend for the two whales (Fig 5).

Generally, the mean breath per minute bin was significantly higher for the second whale (N = 834) (up to 7 breaths per minute bin) than for the first whale (N = 298) (less than 4 breaths per minute bin) (Mann-Whitney U test: z = 15.0, P = 0.0001). However, significant differences were found among phases for the first (Kruskal-Wallis test: $H_c$ = 28.5, df = 5, P = 0.0008) and second whales (Kruskal-Wallis test: $H_c$ = 24.3, df = 4, P < 0.0001) (Fig 6). Particularly, higher breathing rates were found during rescuing operations for the first whale

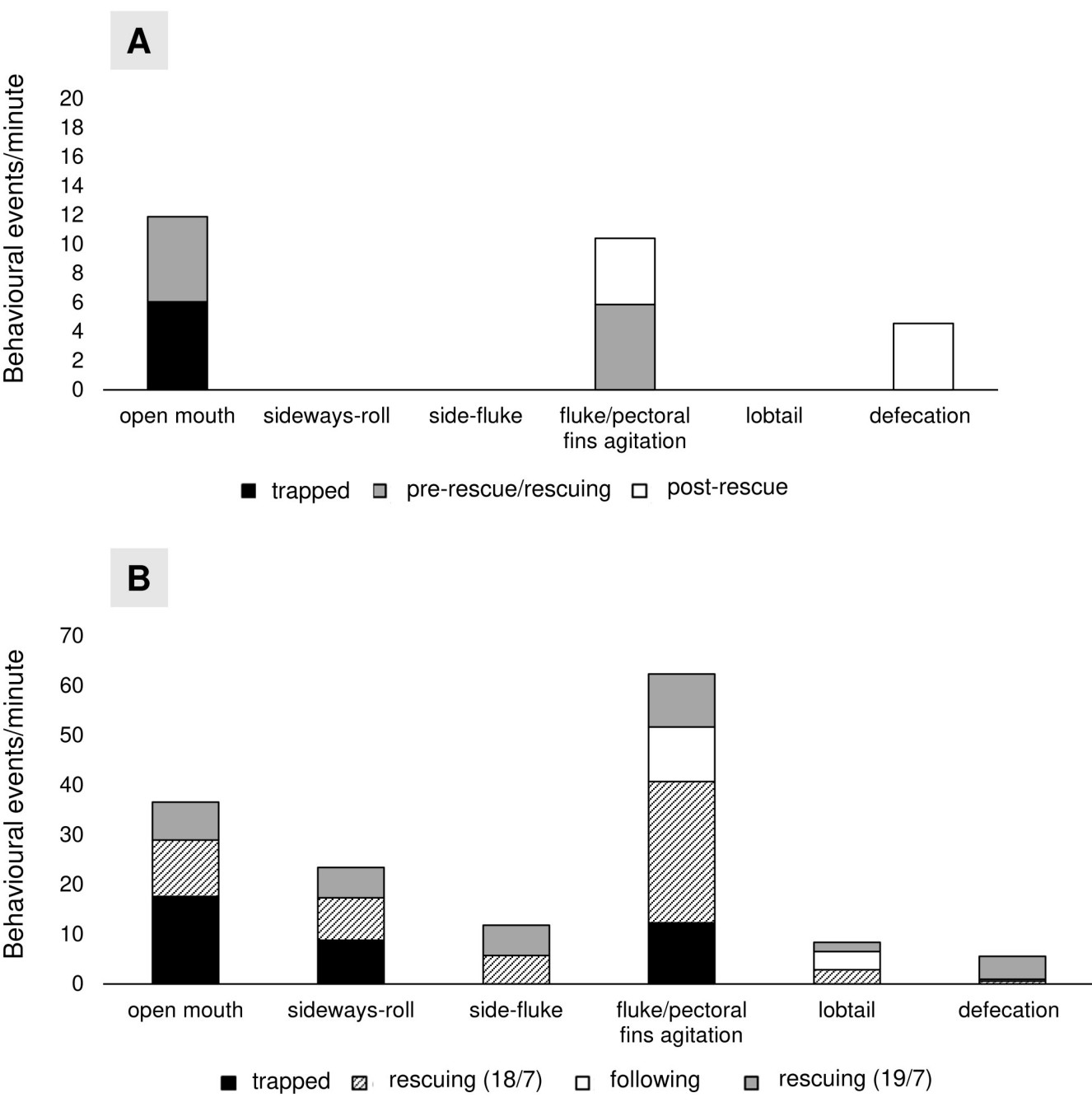

**Fig 4. Rate (%) of each behavioural display in different phases.** A) First whale. B) Second whale.

(Mann-Whitney U test, Bonferroni corrected alpha: P = 0.001) and during entanglement for the second whale (Mann-Whitney U test, Bonferroni corrected alpha: P = 0.001) (Fig 6).

During the "trapped" phase the mean time between two breaths was 6.60 (2.50) minutes, ranging from 2–15 minutes, for the first whale, and it was 2.61 (1.26) minutes, ranging from 1–5 minutes, for the second whale. The first whale never dove during any phase while the average dive time was 13.07 (0.10) minutes for the second whale (calculated over 15 dives) (Fig 7).

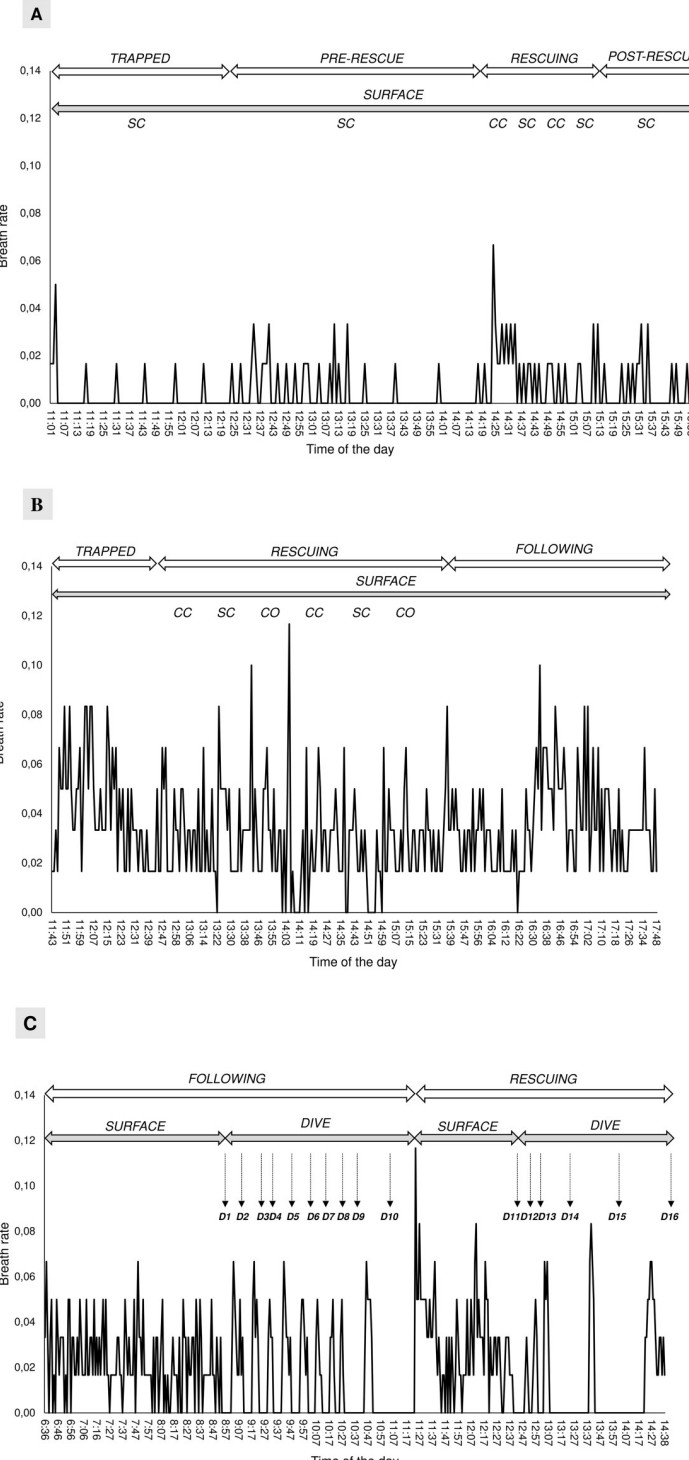

**Fig 5. Breathing rate (breaths per minute bin) trend, surface or diving behaviour, and vocalization recorded.** A) First whale during all phases on 26/06/2020. B) Second whale during the first day (trapped, rescuing and following on 18/07/2020). C) Second whale, second day (rescuing and following on 19/07/2020). SC = single clicks; CC = creaks; CO = codas. Dives are indicated as D followed by a sequential number.

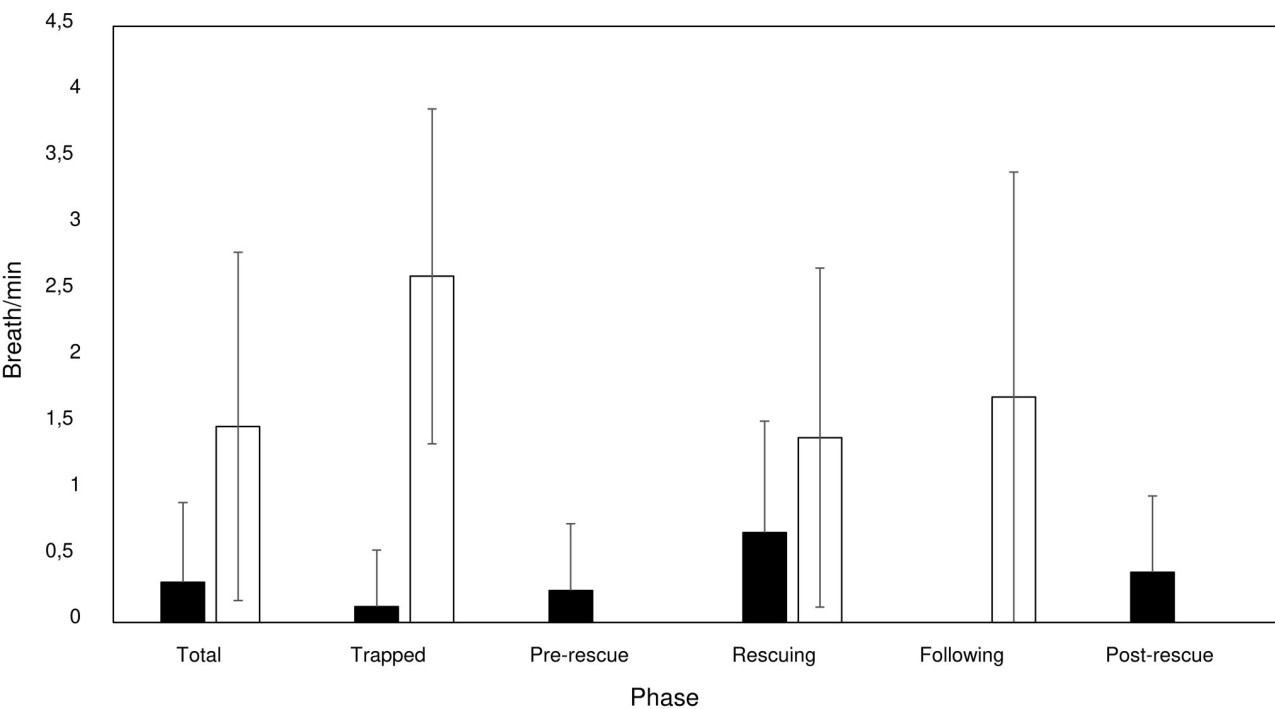

**Fig 6. Mean breath/minute in each phase comparison.** First whale (black) and second whale (white).

D10 was the longest dive performed just before the second attempt of rescuing the second whale. Prior to this long dive, the whale regularly performed other shorter dives (Fig 7). D14 and D15 was the longest dives performed after several rescuing attempts (Fig 7).

### Acoustic behaviour

A total of 34 minutes for the first whale, recorded in three phases (trapped, rescuing, and post-rescue; 26th June), and 22 minutes for the second whale, recorded during rescuing operations (18th July) were analysed. The sequence of vocalizations produced during surface behaviour significantly differed between the two whales. Particularly, the first whale produced primarily single clicks in all phases except for two series of creaks while the rescue operations were underway (Table 1).

The second whale (acoustically recorded only throughout rescue operations), after a first series of creaks (named "creaks 1"), produced few single clicks followed by the first series of codas (named "coda 1"). (Table 1). Then, the second whale started a second series of creaks (named "creaks" 2) followed by a second series of codas (named "coda 2"), composed of just coda_3 and coda_4 (Table 1). The structures of these codas were: 2+1 (coda_3), 1+2+1 (coda_4) and 2+2+1 (coda_5) similar to the ones recorded among the Mediterranean sperm whale population [11, 48].

Additional 1519 minutes (from 19:04 to 06:15) of acoustic recordings collected during the second whale's re-sighting on 5[th] October were analysed separately showing a completely different pattern, i.e., several series of single clicks with ICI from 0.8–1.8 s combined with extensive pauses in click emission (silence periods) [51].

**Single clicks.** For the first whale 612 single clicks were recorded; the lower frequency limit of these clicks ranged between 66.0 Hz and 1885 Hz, while the upper frequency limit ranged

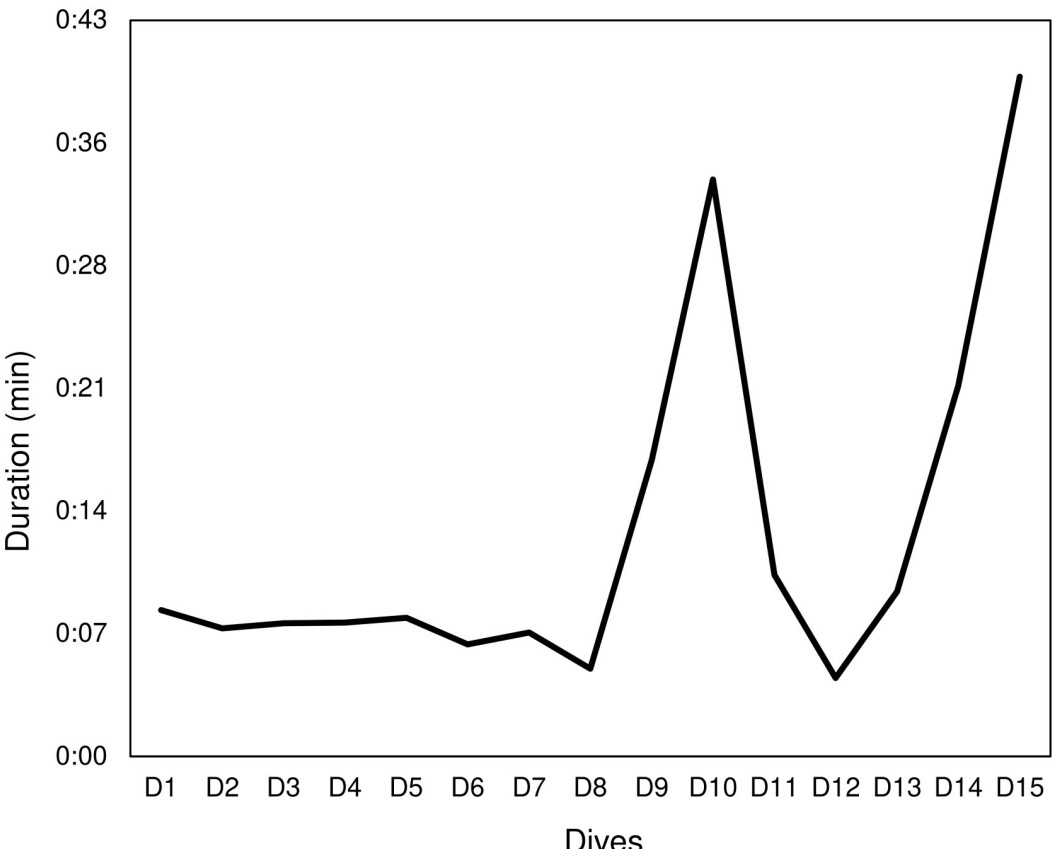

**Fig 7. Duration (minutes) of each dive (from D1-D15) for the second whale.**

from 14.85 kHz to 24.00 kHz. An average click rate of 0.43 clicks per minute was recorded, but the click rate was higher during rescuing operations (0.52) than trapped (0.29) or post-release (0.29) phases. On the contrary, only 12 single clicks were recorded from the second whale during the first day with the lower frequency limit of these clicks ranging between 207.4 Hz and 1653 Hz, while the upper frequency limit ranged from 13.76 kHz to 17.76 kHz. The mean ICI and frequency of single clicks were calculated for each whale (Table 1). For the first whale the mean ICI significantly differed among phases (Kruskal-Wallis test: N = 618, $H_c$ = 65.6, df = 3, P < 0.0001) with higher values during entanglement (N = 322) than in the other phases (N = 296) (Bonferroni corrected alpha: P < 0.0001). However, during rescuing the ICI value resulted significantly higher (more than twice) in the second whale (N = 12) than in the first whale (N = 209) (Mann-Whitney U test: z = 5.9, P < 0.001) (Table 1).

The acoustic dataset collected during second whale re-sight included in total 12 series of single clicks (mean duration (SD): 11.61 (6.79) minutes) with a mean ICI (SD) of 0.91 (0.37) minutes combined with extensive pauses in click emission (silence periods) (mean duration of silence periods (SD): 52.87 (22.16) minutes). The click rate was constant, ranging between 0.55 and 1.25. Creaks and codas were not recorded in this period.

**Creaks.** A total of 63 creaks were recorded, among the two events. As it concerns the first whale, a first series of creaks (creak 1) included five trains and 77 clicks in total, while the second series (creak 2) included only one train composed of a sequence of 10 clicks (Table 1). The numbers of clicks in creaks ranged from 4–23 with a mean value (SD) of 14.5 (8.1) clicks per

**Table 1. Each sound emission type temporal and physical parameters.**

| SINGLE CLICKS | | | | |
|---|---|---|---|---|
| | Phase | Clicks (N) | ICI (s) | Frequency limits (Hz) |
| First whale | Total | 612 | 2.86 (3.99) | 1087 (663)-20629 (3454) |
| | Trapped | 322 | 3.39 (4.89) | 1645 (92)-23877 (214) |
| | Rescuing | 209 | 2.00 (2.51) | 426 (439)-17004 (979) |
| | Post-rescue | 81 | 2.84 (2.17) | 577 (430)-17170 (460) |
| Second whale | Rescuing | 12 | 4.91 (1.27) | 1238 (352)-16441 (1107) |
| CREAKS | | | | |
| | Series | Trains (N) | Train duration (s) | Time interval (s) | Frequency limits (Hz) |
| First whale | 1 | 5 | 0.557 (0.277) | 0.90 (0.58) | 1268 (55)-19964 (75) |
| | 2 | 1 | 0.651 (0.000) | * | 66 (0)-16306 (0) |
| Second whale | 1 | 43 | 0.860 (0.218) | 2.43 (2.95) | 1800 (42)-14653 (4320) |
| | 2 | 14 | 0.746 (0.175) | 1.74 (0.91) | 1766 (40)-16819 (1166) |
| CODAS (second whale) | | | | | |
| Series | Type | Codas (N) | Coda duration (s) | Time interval (s) | Frequency limits (Hz) |
| Coda 1 | Total | 53 | 0.488 (0.025) | 5.06 (2.61) | 1714 (216)-16538 (1334) |
| | Coda 1_3 | 21 | 0.492 (0.026) | * | 1753 (203)-16669 (1687) |
| | Coda 1_4 | 31 | 0.485 (0.025) | * | 1687 (228)-16444 (1079) |
| | Coda 1_5 | 1 | 0.492 (0.000) | * | 1716 (0)-16703 (0) |
| Coda 2 | Total | 10 | 0.462 (0.033) | 6.46 (2.94) | 1306 (61)-16755 (1031) |
| | Coda 2_3 | 3 | 0.476 (0.030) | * | 1313 (69)-17281 (1018) |
| | Coda 2_4 | 7 | 0.455 (0.034) | * | 1303 (62)-16530 (1025) |

Quantity, duration (only for creaks and codas), time interval, and frequency lower and upper limits of each vocalization recorded. Mean (SD) values are reported for all data (Total) and each phase with acoustic data (Trapped, Rescuing and Post-rescue), series (1 and 2) and coda type (with 3, 4 and 5 clicks).

train and 50% of the values are included within the range of 20–23 clicks per train. As it concerns the second whale, the first series of creaks (creak 1) was composed of 43 click trains of 942 clicks in total. The number of clicks in each train ranged from 7 to 37 clicks with the majority of trains (80%) within the range of 19–24 clicks, with a mean number (SD) of 21.9 (6.0) clicks per train. Then the second series of creaks (creak 2) included 14 trains of 306 clicks in total. The number of clicks in each train ranged from 16–32 with 70% within the range of 21–23 clicks and a mean (SD) of 21.9 (4.1) clicks per train.

The mean duration (s) of trains in each series of creaks and the time interval between trains (s) of each series for both whales are reported in Table 1, showing significantly higher train duration (Mann-Whitney U test: $z = 14.1$, $P < 0.0001$) and interval (Mann-Whitney U test: $z = 7.0$, $P < 0.0001$) in the second whale ($N = 57$) than in the first one ($N = 6$). A detailed analysis of each series of creaks was performed (Fig 8) showing significant difference between series for the first whale (Mann-Whitney U test: $N_1 = 77$, $N_2 = 10$, $z = 4.5$, $P = 0.0001$) but not for the second whale (Mann-Whitney U test: $N_1 = 942$, $N_2 = 306$, $z = 2.4$, $P = 0.02$).

Although several factors might have influenced whales' acoustics emissions (i.e. timing at which whales had been entangled, number of boats present in the area, physical status, natural inter-individual variability, etc.), it is noted that creaks occurred only during the "rescuing" phase when the whales showed different behavioural displays related to stressful and traumatic conditions. In particular, for the first whale the first series of creaks was recorded at the beginning of rescuing operations, while the second series when half of the net was removed from its tail. Vigorous fluke movements and mouth opening were associated with creaks. For the

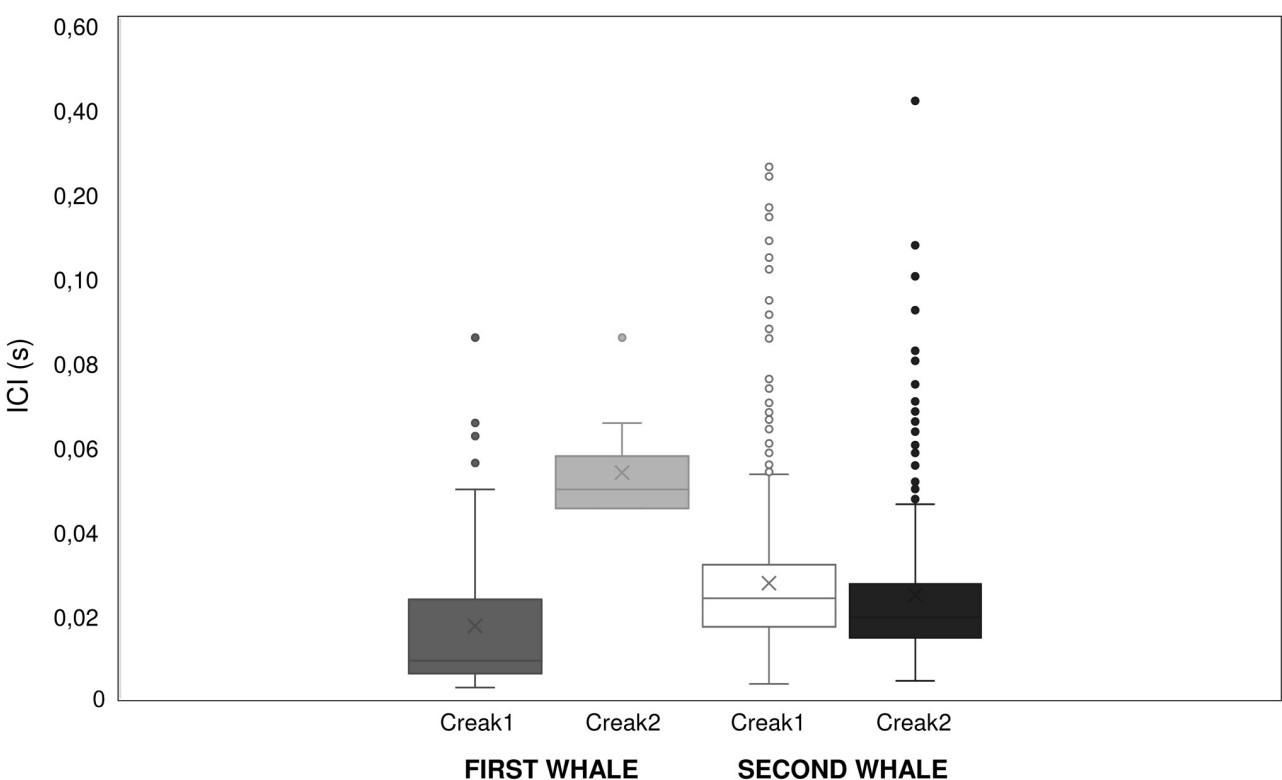

**Fig 8. ICI (s) for each creak series in both whales.**

second whale both creaks series were heard during the first day when several people (rescue divers and filmmakers) swam in the proximity of the animal; moreover, on the first day, the rescue team tried more times to remove the net with boat hooks, which got entangled in the netting. Agitation of fluke and pectoral fins, frequently opening the mouth and sideway rolling, and side fluking was also observed in association with creaks.

**Codas.** Codas were recorded for the second whale only. In total, sixty-three codas were detected, distributed in two different sequences (coda 1 and coda 2) (Table 1). A mean coda occurrence rate of 0.18 codas per second was recorded, but the coda 2 mean rate was slightly lower (0.16 codas per second) than coda 1 (0.19 codas per second). The inter-coda interval ranged from 2.8 to 21.6 s. The duration of codas ranged from 412-561 ms but 85% lasted between 440 and 500 ms with a mean duration of 484 (28) ms. Three coda types, containing 3, 4, and 5 clicks, were found and classified in accordance with Weilgart and Whitehead [52] (Fig 9).

Among all codas, 3-click codas (38% of codas; type 2+1) and 4-click codas (60% of codas; type 1+2+1) were found to be the most common, with the exception of a single coda formed by 5-clicks (type 1+2+2) occurred just once, while operators tried to disentangle the net using the first boat hook. A detailed analysis within codas pattern showed great variability in ICI duration, ranging between 2–300 ms (Fig 9). The two series of codas were mainly heard during the first day, when the whale, once the first part of the net was removed, started to move away from the rescue team, showing an active behaviour, vigorously moving its fluke and pectoral fins, frequently opening its mouth and sideway rolling or side fluking and defecating.

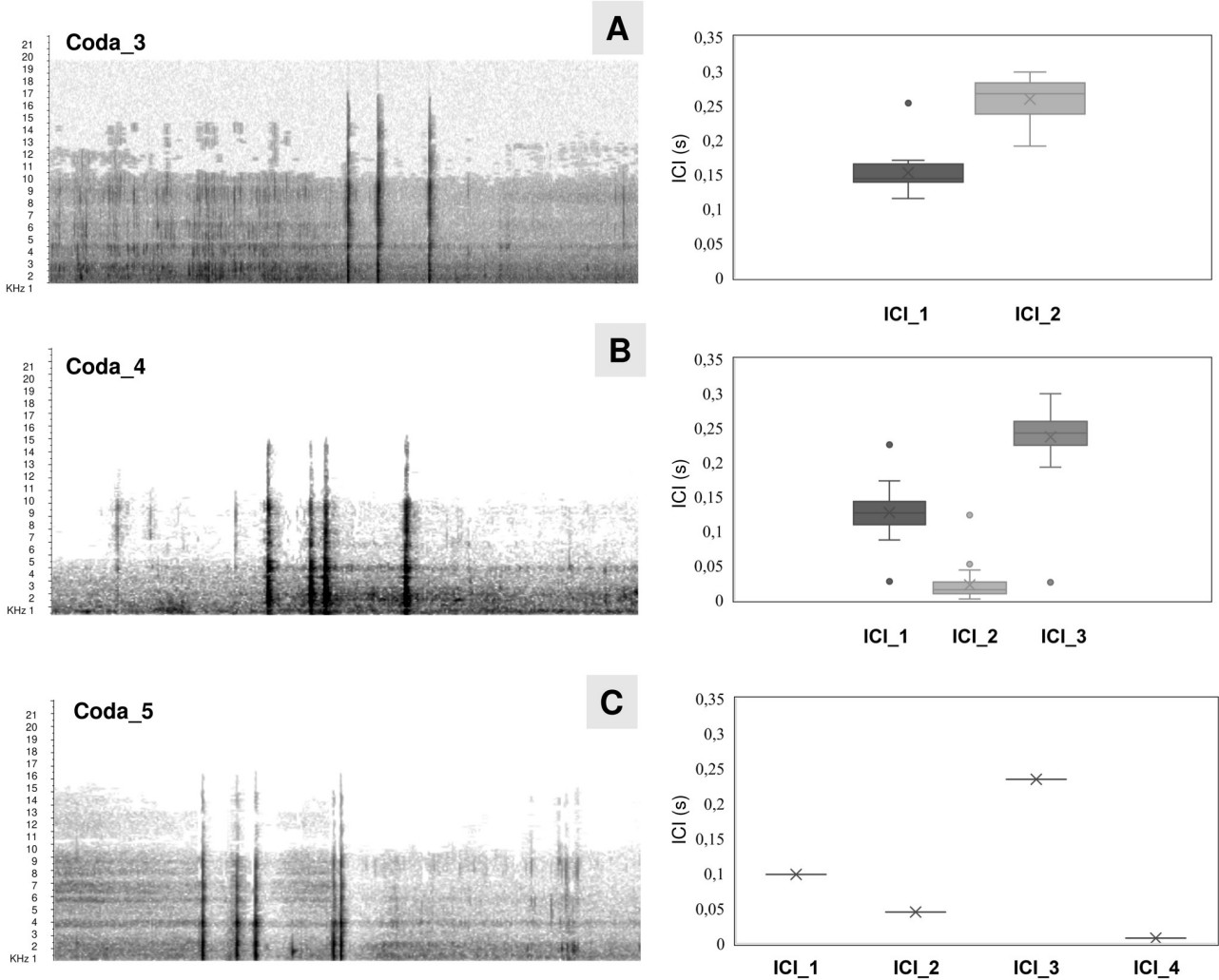

**Fig 9. Spectrograms of coda types and related inner-coda ICIs (s).** A) coda_3 (3-clicks). B) coda_4 (4-clicks). C) coda_5 (5-clicks). ICI_1 = ICI between the first and the second click; ICI_2 = ICI between the second and the third click; ICI_3 = ICI between the third and the fourth click; ICI_4 = ICI between the fourth and the fifth click.

## Discussion

This study provides new data on behavioural pattern, including acoustic behaviour, displayed by sperm whales under particularly stressful conditions (i.e. entangled in illegal driftnet targeting large pelagic species [53–57]) and compared these resulting observations to the existing literature about the acoustic behaviour of sperm whales in natural conditions [39, 40] and under similar or other (i.e. human-made noise) stressful contexts [11, 41]. Combining behavioural, respiratory, and acoustic data allowed us to gain a wider understanding on how sperm whales may react to entanglement and, consequently, to better manage rescuing operations, in order to reduce additional stress or accidental injuries to the animal.

Despite similarities between the two situations, two different reactions were documented. The disentanglement operations on the first whale lasted less than an hour and successfully freed the animal. On the other hand, the driftnet was completely removed from the body of the second whale, but not from the tail. The animal got nervous and started to swim away

from the rescuers, who continued to search for it the next day, even after its disappearance. The different whales' responses may be related to several factors, such as the individual's behaviour, its health status, and the specific reaction to disentangling operations and to the rescue operations themselves (different number of vessels and operators in the rescue area).

The behavioural displays observed in this study have been previously reported in other cetacean species, primarily during aggressive/agonistic attacks or in stressful circumstances [58]. Our data are not enough to confirm such behaviours, which could be stereotyped postures of the species [59, 60] or behavioural modes adopted during particularly stressful conditions expressed in the proximity of an imminent danger [11]. Moreover, it is not excluded that these behaviours may be accentuated during traumatic events, occurring more frequently in stressful circumstances than under normal conditions [11]. In this study, only the second whale showed sideway rolling and side fluking, both behavioural displays which require a greater energy demand for a whale under traumatic conditions. This leads to the hypothesis that the entanglement of this whale may be recent compared to the first whale, which was found generally in poor health condition.

Our data suggest that the breathing pattern may be a good indicator of stress during entanglement events. The first whale showed significantly higher breathing rates during rescuing operations than in the other phases. Moreover, the second whale showed higher breathing rates than the first one. Considering the massive human presence around the second whale, this is consistent with the hypothesis that external disturbances may cause additional stress to the animal. Particularly, frequent and extended breaths at the surface are indicators of proximity with an upcoming dive, as these breaths enhance the diving time and the opportunity to execute longer dives in depth [61, 62]. On the contrary, when the breathing rate is less frequent, the dive is likely to have ended and the whale is resting and preparing for the next dive [63]. The duration of dive is known to be partially physiologically limited, but in most cases, it is under behavioural control [64, 65]. Ventilation is the result of respiratory frequency and tidal volume, and both parameters usually increase with the longer dive durations [66]. This type of respiratory alteration observed as increased ventilation has been reported for a variety of mammals [67, 68] and cetaceans [69], but never for rescuing operations of entangled sperm whales. Increased ventilation is necessary because of the oxygen deficit and accumulation of carbon dioxide acquired during submergence [70]. The rapid replacement of oxygen stores throughout the body is further facilitated by an increased heart rate during these surface intervals [71, 72]. The dive duration trend of second whale suggests that an increased ventilation might restore the high diving costs of the residual embedded net [73].

In this study, the acoustic pattern was analysed according to other studies on marine mammals [34, 74–78]. Although the entanglement conditions were similar, the two whales presented a different acoustic pattern. The first whale produced primarily single clicks (99.0%) in all phases except for two series of creaks (creak 1: 0.8%, creak 2: 0.2%) when rescue operations were in progress. The second whale after the first series of creaks (32.6%) produced few clicks (9.1%) followed by a first series of codas (40.2%) and a second series of creaks (10.6%) and codas (7.6%).

Particularly, in this study, the term 'single clicks' is used to indicate those sounds which resembled regular echolocation clicks ("usual clicks") associated primarily with an echolocation-based foraging [79] but with some substantial differences. Firstly, click analysis shows a high average ICI (2.86 and 4.91s for first and second whales, respectively), consequently average click rates were very low (0.4 and 0.2 sec$^{-1}$ for first and second whales, respectively) in comparison with typical echolocation clicks used by whales in search of food [51, 79]. Jaquet et al. [38] described "surface clicks" as "vocalizations had a long inter-click interval (5 to 7 s on average, also called 'slow clicks') and sounded very metallic" ("clangs" according to Gordon [80]).

Although the situation described in this study was atypical, plausible hypotheses might be that these clicks were used to echolocate the position of the rescuers [46], or also to keep contact with nearby whales [81]. However, for the second whale, the clicks emitted at the surface, did not fit perfectly with the description of "surface clicks" [38]. Commonly, "usual clicks" are usually produced in prolonged bouts interspersed with buzzes [79], which is true for the first whale (it constantly emitted clicks with slight fluctuations of the inter-click interval in accordance with the activity level, Table 1), but it is not clear for the second whale, which emitted just a 12-click train at a very slow rate between a long creak sequence and coda sequence, both triggered during high activity situations. Previous studies have shown that the click rate can be subjected to variations related to activity, group size, and physical condition [38, 44, 46]. The low click rates recorded for the first whale could be ascribed to a general state of weakness, due to the excessive suffering condition caused by the entanglement. Accordingly, when the second whale has been re-sighted, the acoustic recording included only usual foraging clicks (click rate: 50.3 (17.6) min$^{-1}$, N = 96; ICI range 0.86–1.83s, mean ICI: 0.91 (0.37) s), which are typically interspersed with periods of silence (~52 min average) [34], indicating its ability to dive (at which depth is unknown) and likely to forage, despite the residual net wrapped on the tail. A comparison with behavioural and breathing pattern data of this encounter was not performed because the whale was encountered just before sunset.

Creaks are a wide kind of vocalizations that include two main categories: long and short-lasting creaks [34, 45]. The first type is mainly associated with deep dives echolocation, especially in prey detection and targeting [34, 38, 45, 79, 80, 82–85]; while the second type includes "coda-creaks", "chirrups" or "rapid-click" [47, 80] and is emitted in a social context for scanning other whales while lying at the surface [34]. In this study, creaks were recorded for both whales and shared similar acoustics parameters, of which some were consistent with previous literature. The ICI values, for instance, ranged between 5–100 ms like reported by Pace et al. [11] for whales in entanglement conditions. Additionally, the structure and the number of clicks contained in trains (range 10–37 clicks) is consistent with Goold's "chirrups" (range 10–50 clicks) [47]. On the other hand, others vocalization characteristics were definitely different, such as duration (range 0.7–139 s) and click rates (21.4 sec$^{-1}$ for the first whale and 30.9 sec$^{-1}$ for the second whale), showing lower values than usual social creaks [47]. Despite those creaks had different occurrence rates (first whale: 1% of the total analysed emissions (N = 618); second whale: 43% of the total analysed emissions (N = 132)), those sounds were emitted only during the "rescuing" phase. In particular, when rescuers were operating around the whale by cutting or pulling the net off, suggesting that those sounds might be related to an extreme uneasiness. Furthermore, those sounds were associated with high breathing rates and distress-related behavioural displays. In nature, several animals emit sounds in the proximity of imminent danger or during traumatic events and stressful circumstances [86], like trapping [87–90], or, as observed in bottlenose dolphins (*Tursiops truncatus*) to elicit aid from another animal by emitting repetitive whistle sequences [91]. Another study [92] on bottlenose dolphins shows that the occurrence of whistle production was positively correlated with the occurrence of the supporting behaviours which the injured dolphins received from other individuals. Finally, Tellechea et al. [93] reported the sound behaviour of a stranded humpback whale (*Megaptera novaeangliae*) which emitted atypical "grunts", normally associated with foraging, with a duration similar to the creaks found in this study (duration range 0.56–0.86s). Our data are not sufficient to assume the true function of these sound emissions although it is not excluded that their ICI and duration values could be related to the physical/psychological status of the animal or also a natural-intra-individual variability.

Codas are known to be involved in communication between sperm whales [34], and different coda repertoires may occur among populations ("vocal clans") [94, 95]. The codas

repertoire found in this study showed some similarities, mainly in its structure and duration with codas reported in other studies [48, 96–99]. Analogously to that reported in the Mediterranean Sea [48], the 3-clicks coda type (2+1) (38.1%) has been found, fitting in the description of a typical "2+1" coda family [11], with a coda-inner click interval ranging from 0.15 to 0.28 s. The occurrence of (2+1)-click type has been also described off the Balearic Islands [97] and in the Tyrrhenian Sea [96], even though it is believed to be more common among socializing whales than in solitary individuals [11]. Besides, this coda type was already found in whales in stressful conditions [11]. However, the recorded repertoire was dominated by a 4-clicks (1+2 +1) coda type (60.3%), already reported in the Mediterranean Sea [11, 48, 99]. A single coda train of 5-click (1+2+2) has been also detected, previously documented only once for a group of sperm whales (group size > 3) in Greece (Aegean Sea) [48, 94, 100]. Finally, coda durations (0.4–0.5 s; overall mean duration 470 ms) were also similar to those reported in socializing whales off the Tyrrhenian Sea (range 0.4–1.2 s; overall mean duration 908 ms) or for entangled whales (range 0.2–5 s; overall mean duration 398 ms). On the other hand, the most important difference found in this coda repertoire is the values of the ICI within codas, especially for 4-clicks and 5-clicks types, which were more similar to creaks' (ICI < 0.1 s) than Mediterranean socializing and entangled whales' codas (ICI: 0.1–0.5 s). It is not excluded that the observed ICI differences may be related to the low coda sample collected in this study, but the co-occurrence of a short creak train between codas and the absence of other whales nearby, suggest that those vocalizations may be emitted for a different purpose than socializing. Among the probable hypothesis about the behavioural significance of these codas, the more reliable can be found in a particular category of codas, called "alarm codas", reported by several authors [41, 101]. It has been observed that those vocalizations occurred when solitary whales or non-socializing groups are distressed by human or predator (killer whale, *Orcinus orca*)' presence, and in association with altered surface behaviour. Accordingly, our codas have been produced by a non-socializing whale in a situation of disturbance and imminent danger, together with atypical surface behavioural displays ("opening of mouth", "agitation of fluke and pectoral fins", "sideway rolling", "side fluking" and "defecating"). However, only few data were analysed and this hypothetical association between behaviour and acoustics needs further data and observations to be confirmed.

## Recommendations

Past experiences with entangled cetaceans have shown that a clear and comprehensive set of guidelines should be followed to decrease the risk of mortality and/or further injury or stress, thus increasing the probabilities of post-release survival [50]. Experience gained from the present study shows that a high level of competence and preparedness, which includes being prepared with the right equipment, has a significant positive impact on successful disentanglement operations [50]. Before starting the rescue operations, a preliminary assessment of the general condition of the animal is recommended, to identify priorities and define appropriate and timely responses, and therefore desirable results [50]. Furthermore, it is essential to minimize the possible stress caused by the presence of boats and/or operators not involved in the disentanglement process [50]. However, psychological and physical stress in entangled whales is a key issue to keep under control, to minimize unexpected reactions during rescue operations, that could pose a risk to both whale (who may accidentally get injured) and operators (whose safety could be put at risk by sudden movements of a large animal). But this level of stress is difficult to detect and measure, even if it should be treated, reducing it wherever possible. In this paper behavioural, breathing and acoustic recording has proven to be a useful tool and a non-invasive way to continuously monitor the physical and

psychological condition of entangled sperm whales, suggesting that these data should always be collected to continuously monitor whale status during the rescue operations.

## Acknowledgments

We thank all people and organizations who collaborated on this work. Francesco Principale and the Coast Guard of Lipari, Catania, Messina, Milazzo, and Napoli for the incredible effort in these two entanglement cases to save the two individuals of sperm whales. Moreover, we thank Lipari Diving and Muciara Diving for the collaboration in rescuing operations. Dr. Clara Monaco for reporting us the second whale entanglement. We also thanks Sea Shepherd Italy for the logistic support during the night monitoring sessions. Last but not least, we thank the volunteers and students of Filicudi WildLife Conservation who assisted during the monitoring sessions supporting the team in the collection of data. This work is part of the activities of a European Life Project named Life Delfi aimed to reduce negative interactions between fisheries and cetaceans in different Mediterranean areas. No invasive methods have been used in this study; hence no ethics statement is required.

## Author Contributions

**Conceptualization:** Monica Francesca Blasi.

**Data curation:** Valentina Caserta, Chiara Bruno, Perla Salzeri.

**Formal analysis:** Monica Francesca Blasi, Valentina Caserta, Chiara Bruno.

**Investigation:** Monica Francesca Blasi.

**Methodology:** Monica Francesca Blasi, Valentina Caserta.

**Software:** Valentina Caserta, Perla Salzeri.

**Supervision:** Monica Francesca Blasi.

**Validation:** Monica Francesca Blasi, Perla Salzeri.

**Writing – original draft:** Monica Francesca Blasi, Valentina Caserta, Chiara Bruno, Perla Salzeri, Agata Irene Di Paola.

**Writing – review & editing:** Monica Francesca Blasi, Valentina Caserta, Alessandro Lucchetti.

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
