## [Decision Letter · Decision Letter 0]

12 Feb 2021

PONE-D-20-40631

Behaviour and acoustic vocalizations of two sperm whales (Physeter macrocephalus) entangled in illegal driftnets in the Mediterranean Sea

PLOS ONE

Dear Dr. Blasi,

Thank you for submitting your manuscript to PLOS ONE. After careful consideration, we feel that it has merit but does not fully meet PLOS ONE’s publication criteria as it currently stands. Therefore, we invite you to submit a revised version of the manuscript that addresses the points raised during the review process.

We look forward to receiving your revised manuscript.

Kind regards,

Patrick J. O. Miller

Academic Editor

PLOS ONE

Additional Editor Comments:

Your manuscript received mixed reviews, and clearly a substantial edit and improvement is needed. Please revise addressing all of the comments of the reviewers, paying particular attention to reduce or more fully justify speculative comments in the discussion as pointed out by reviewer 3.

Journal Requirements:

2. We note that Figure 1 in your submission contain map images which may be copyrighted. All PLOS content is published under the Creative Commons Attribution License (CC BY 4.0), which means that the manuscript, images, and Supporting Information files will be freely available online, and any third party is permitted to access, download, copy, distribute, and use these materials in any way, even commercially, with proper attribution. For these reasons, we cannot publish previously copyrighted maps or satellite images created using proprietary data, such as Google software (Google Maps, Street View, and Earth). For more information, see our copyright guidelines: http://journals.plos.org/plosone/s/licenses-and-copyright.

2.1.    You may seek permission from the original copyright holder of Figure 1 to publish the content specifically under the CC BY 4.0 license. 

2.2.    If you are unable to obtain permission from the original copyright holder to publish these figures under the CC BY 4.0 license or if the copyright holder’s requirements are incompatible with the CC BY 4.0 license, please either i) remove the figure or ii) supply a replacement figure that complies with the CC BY 4.0 license. Please check copyright information on all replacement figures and update the figure caption with source information. If applicable, please specify in the figure caption text when a figure is similar but not identical to the original image and is therefore for illustrative purposes only.

Reviewers' comments:

Reviewer's Responses to Questions

**Comments to the Author**

1. Is the manuscript technically sound, and do the data support the conclusions?

Reviewer #1: Yes

Reviewer #2: No

Reviewer #3: Partly

2. Has the statistical analysis been performed appropriately and rigorously? 

Reviewer #1: Yes

Reviewer #2: No

Reviewer #3: I Don't Know

3. Have the authors made all data underlying the findings in their manuscript fully available?

Reviewer #1: Yes

Reviewer #2: Yes

Reviewer #3: No

4. Is the manuscript presented in an intelligible fashion and written in standard English?

Reviewer #1: Yes

Reviewer #2: Yes

Reviewer #3: No

5. Review Comments to the Author

Reviewer #1: Review PlosOne D-20-40631

This paper presents an interesting description of two net entangling events that occurred to sperm whales around the Aeolian Islands in the lower Tyrrhenian Sea. The authors describe the disentangling operations and the associated behaviour of the two animals during the rescue operation. The paper is pretty descriptive, and scientific results are limited to a description of occurred behaviours, rather than a scientific interpretation; however, the paper depicts the problem of illegal drifnets and gives cues and recommendations about how to face such entangling events and related rescue operations. Considering this, I recommend the publication with a minor revision to correct some inconsistencies and ambiguities in the acoustic behaviour description. Overall, it is a very interesting lecture on an unusual and poorly known event.

124: please add more information on recording: depth of the hydrophone ? sampling rate of the recorder ?

127: measure rather than “extrapolate”

127: frequency parameters rather than just “frequency” if you mean measures of the frequency range of the vocalizations; otherwise if you mean “frequency of occurrence” please explain better.

243: please briefly explain the structure/pattern of the codas with references to available literature for Med codas (to be discussed in later in the paper)

247: (12 spectrograms) ? what do you mean ?

251: the description of frequency limits is not clear. Use the expression “lower frequency limit” and “upper frequency limit”. Or just refer to the "frequency range". Also explain how the frequency range has been calculated, e.g. at -XX dB ref max amplitude in the spectrum ?

253: “click frequency rate” is incorrect: to avoid confusion don’t use the term “frequency” when you write about “rate”

256: same as 251

258: “frequency” do you mean “frequency range” ?

264, 277, 279: it is not clear the meaning of the number in parentheses

301: per second … for second … please correct

Discussion

interesting to note that your codas are different from the classical 3+1 coda reported in literature as the most common in the Med. However the 2+1 has been already reported in the Med. Also interesting to note the production of different codas by the same individual.

428: Pace et al. in parentheses rather than referenced by a number.

Biblio:

18.

For the Italian strandings database it is also worth to mention this paper, unfortunately in Italian, but the only available to describe the data bank

Pavan G., Bernuzzi E., Cozzi B., Podesta M., 2013. La rete nazionale di monitoraggio degli

spiaggiamenti di mammiferi marini. Atti 44 Congresso SIBM. Biol. Mar. Mediterr. (2013), 20:

262-263.

Reviewer #2: General comment:

The authors present two rescue operations of sperm whales entangled in driftnets in the Aeolian Archipelago. Behavioral data of both sperm whales were collected over the rescue operations. This study is not a research paper where you expect a clear biological question and associated reproductible experimental design with sufficient sample size to answer your hypothesis. This work corresponds to a case report paper which provides informative and useful material for futur similar rescue cases and for supporting regulations related to illegal use of driftnets at risk for whales’ entanglements. I recognize the great effort that was accomplished here with both the rescue process and the collection of behavioral data. This represents an obviously important piece of information, from a conservation point of view, and also for enriching our knowledge on behavioral features displayed by sperm whales under particularly stressful contexts. The manuscript contains data that deserve to be spread out into the scientific community, however, it does meet the criteria for a publication in PLOS ONE (2 report cases, and see specific comments). May I suggest a more appropriate publication of this work as a report paper in "Journal of Cetacean Research and Management" or as in "Aquatic Mammals". I list below some specific comments and suggestions that I hope will be useful to improve some aspects of the manuscript.

Specific comments :

1) Objectives and focus of the paper

In the introduction, the authors provide a rich documentation on the issue of illegal use of driftnet, highliting well the conservation problem for the species. However, a followed clear objective is missing.

I found attempts to state objectives/hypothesis to be tested here :

- in the introduction, L. 79-83 (“The study of the phases following the rescue operations can provide … increasing the chances of post-release survival.”)

- in the discussion, L 324-327 (“Combining behavioural, ... to reduce additional stress or accidental injuries during rescue operations.”).

L. 482. What would yo do during a rescue operation if acoustical features associated to an increased-stress would occur? I don’t think that such monitoring sound production would make a rescue team to decide to interrupt the rescue operation.

There is not enough material to accurately test whether acoustics can allow monitoring stress level, neither to estimate whether the protocol allowed for more or less chances of post-release survival. I recommend to maintain the focus on the actual key outcomes of the paper: i) the rich description of behavioral displays (including acoustic data) exhibited by sperm whales in particularly stressfull conditions (L. 77-78, L. 321-322), and ii) the list of valuable recommendations based on this experience study, that should be useful for potential futur rescue operations (§L. 470).

The descriptive analysis of the behavioral change over the rescue phases is interesting to document (L. 208 and L. 253 « In particular, a higher breath rate was recorded during ongoing rescue operations for both whales”, “…the click rate was higher during ongoing rescue operations”). However, to my opinion, they do not represent the most interesting findings; the paper would have overall more value and would be also attractive for a broader reader community by shaping/accentuating the focus into: providing new data on the behavioral pattern (especially on acoustic behavior) displayed by sperm whales under particularly stressfull conditions (entangled, rescue operation) and comparing it to the existing litterature, i.e. ref. on the typical natural behavior (e.g. how is it common to produce creacks for juvenile sperm whales there and for the species in general…etc), and ref. on the behavioral pattern exhibited in other stressfull contexts such as in response to an increased predation risk or other types of disturbances (e.g. man-made noise) for which there is an existing literature.

2) I don’t see the relevance to statistically compare the behavior of both whales (e.g. L. 282: comparison of number of creaks between both whales): the sample size is critically low (n=2), there is no data on the whales’ baseline behavior to assess for a natural inter-individual variability, the timing at which whales had been entangled when discovered is unknown, and there were additional factors (e.g. inconsistencies of the rescue operation phases across whales, external factors like number of boats present in the area etc) that might also have influenced the whales’ behavior.

Given this highly variable context and factors, I would not focus on comparing both whales in the results section but rather would keep this in a descriptive way for the discussion, with proposing some explanation for such differences (as done in L. 366).

3) comparison across the rescue phases.

The 2 by 2 comparisons between the different phases were made (e.g. L117-118; L.217). Did the authors account for multiple comparisons (e.g. Bonferroni correction).

If the authors aim to make comparison (even an descriptive one) between whales through the different phases, it would help to provide more details of each phase and clarify the level of consistency of the phases between the whales.

-e.g. L. 103, and Figure 4: “monitoring” phase for whale #2 is actually a post-release phase with presence of operators tracking the whale in order to attempt to remove a remaining piece of net. I suggest to delete the term “monitoring” phase and instead have the “post-release” phase with (for whale #2) or without (for whale #1) tracking.

-L. 100: it is only said here that during pre-release period the “operators approached the whale” but there was also a rope that was attached to the tail as mentioned later in the text (Fig 2, and L. 144-146).

4) interpretation.

Figure 4: Do you have reference to support that those behavioral displays are stress-related ? For the increased breath rate, your justification supporting the stress state is more furnished and thus more convincing (L345-364) that for the other behavioral parameters (acoustics, and surface displays).

L. 27: “ …it collaborated during rescue operations showing only few behavioural evidences of stressful condition.”

Getting nearly quiet and hardly moving could actually indicate an extreme degree of stress. So I would not restrict the “stress-related” indicators to the agitation and movements displayed by the Whale number 2.

There are a considerable list of references but the findings are not well shown in the light of them (e.g. with what we know on the acoustic behaviour of sperm whales). I suggest to clearly present what is know on the species regarding the typical natural breath rate, and the known stressfull behavioral contexts for which particular displays such as codas were shown to be produced e.g. anti-predatory responses, etc.); this would consolidate the interpretation of the behavioral data analysis.

L. 336-337- (ref #58) : this is a ref of dolphins in zoo-captivity that showed agonistic interactions with conspecifics. Could you detail how it supports (e.g. common aspects) what was observed in the sperm whales’ behavioral displays here?

- L. 424. and L. 449. In addition to ref 41, it could be mentioned that codas can be produced in a context of anti-predator behavior which represent a particularly stressfull context for the whales (e.g. in Curé et al. 2013. Responses of male sperm whales to killer whale sounds: implications for anti-predator strategies. Sci Rep. 3:1579. doi: 10.1038/srep01579.)

- See also this additional ref for the discussion part on clicks: Tønnesen et al. 2020. The long-range echo scene of the sperm whale biosonar. Biol Lett. 16(8):20200134. doi: 10.1098/rsbl.2020.0134.

Other comments:

- L. 75, 78: “Acoustic vocalizations”. Vocalizations are “acoustics” so you can delete it. I’d actually rather talk about “acoustic behavior of” or “sounds produced”.

- L. 87 and L. 93: the authors mention the first whale as a “10m juvenile male sperm whale”. Does the gender and estimated age class can be also mentioned for the second 9m whale? Also, a more furnished description of the physical aspects of the two animals (body condition) would feed further potential explanations for the different behaviors between whales.

- Figure 4 : Why to separate the pre-release and ongoing phases if data are pooled together at the end?

- L. 367: Data do not suggest that. These are only hypotheses proposed to explain why differences are observed.

Reviewer #3: Behaviour and acoustic vocalizations of two sperm whales (Physeter macrocephalus) entangled in illegal driftnets in the Mediterranean Sea Review of PONE-D-20-40631

This paper reports some behavioural observation of two sperm whales during operations to attempt to disentangle them from driftnet fishing gear. The observations are intersting and the lessons learned could potentially be applied to other similar situations. Unfortunately the manuscript as presented is not suitable for publication. The main problems are (1) a lack of clarity and precision in the descriptions of data collection, processing, and analysis, especially a lack of clarity over which hypothesis tests were specified a priori and why and which were conducted post-hoc having viewed the data and thus carrying much less weight. The distinction must be made clear in any revision. (2) A series of speculative and unsupported claims about vocalisation function in the discussion which are really not supported at all by the data collected, which of its nature simply is not fit to test these kinds of ideas being entirely opportunistic and observational and (3) the need to improve the written presentation with the help of a colleague with full professional proficiency in scientific English.

An ethics statement should be provided. No justification is provided for restrictions to data availability - this does not seem commensurate with contemporary scientific expectations - I do not accept that 'all relevant data are within the manuscript'. In my eyes, this compromises the integrity of the work.

L19 under which jurisdiction? Italy? EU?

L23 it is an opportunity to study 'whales in extremely stressful conditions' but why more broadly is that important?

L31 is 'single clicks' what are called regular clicks in the rest of the literature? Better to keep consistent

L33 'associating' is vague

L35 I do not support the assertion of 'alarm' and 'distress' in these vocalisations - describe the form but extra evidence is needed to ascribe function

L47 there are more recent IUCN assessments of this threat - it is changing and better to reflect the most recent assessments see Notarbartolo et alarm

L55 'national database' - which nation!? Are stranding records from 1714 really thought comparable to those in the last two decades? Maybe revise this to a more pertinent timescale.

L57 why is the location relevant? isn't it the cause we are more concerned about?

L69 why are vocalisations not a behavioural observation? Also 'acoustic vocalisation' is redundant - there aren't any non-acoustic vocalisations right?

L76 'immediately after release, and in the hours-days thereafter' the distinction between these two categories is not clear

L77-78 see point above from abstract - yes a great opportunity but to what wider relevance?

L80 '(i.e. acoustic parameters' this is not a method

L82-83 this level of effort is surely only justified if the fate of one or two individuals over a 5-10 yesr period will make a population level difference - I don't think Med sperm whales are that badly off yet. Perhaps the argument for preparedness could be used - not a major problem yet but could become vital if population trajectories continue downward

L86 I think 10:01 UTC+1 should be the standard time format throughout but the journal may have style rules

L87-94 it would be interesting to know if the nets appeared superficially similar suggesting they sourced from the same fishing operation or different

L93 no sex information for the second whale?

L96-97 it is not neccesary to describe operations down to the level of plotting points on a map - if the GIS was used to produce a particular piece of data from a geographic database, fair enough.

L100-101 what was 'pre-release' only relevant for the first whale? What was the rationale for a distinctive timeline phase? More information is needed on why different timelines were used for each whale.

L108 I think this is best written as 'counting breaths per minute' since focal observations sampling at one-minute intervals strongly suggests point sampling which I don't think it what the authors did since they refer to breathes per minute later

L113 this is confusing because the activities listed are events - the animal performed a specific behaviour like side-roll or spy hop - so how are 'percentage of each activity' calculated specifically?

L118-119 no pairwise follow up in the case of a significant KW result?

L120-122 what is the theoretical justification for expecting a monotonic if non-linear correlation between time of day and length of dive? Was this hypothesis established a priori or tested after viewing the data? The difference matters!

L123 this sentence is also true if you remove 'stressed'...

L125 give at least the hydrophone +/-3dB frequency response here and give more detail on what the 'wideband solid-state recorder' was - sampling rate? Bit depth? Make/model?

L127 I don't think 'extrapolate' is the right word here.

L128 'single clicks' is not something I recognise from the literature - do the authors mean 'regular clicks' or 'slow clicks' as characteristically produced by males?

L169 what is a 'lighting boa'? A lit mark of some kind?

L171 I think perhaps 'long dives' rather than 'deep' would be more precise here?

L173-174 this is kind of vague and speculative - what is a 'great exploration effort'? Better to say 'X boats, Y planes and Z individuals'

L198 this is exactly the same information that started the previous section - avoid this repetition

L210-211 I am unclear what is being tested here - a direct comparison is suggestd by the text specifically between whale 1 vs whale 2, in which case only pairwise tests are needed? So what was the multiple sample KW test used for? Please provide more clarity here.

L220 pretty sure this should be 'breath/minute' - if you count the breathes in a minute then divided by 60 then you get breath/second... please check and clarify

L216-217 again I am unclear how these tests are working - the KW test gives you a test of whether all the samples come from the same population but doesn't specify which categories vary, for that you need post-hoc pairwise tests

L220 this section bascially confirms to me that the test of the dive time vs time of day was a post-hoc hypothesis test and should be labelled as such (e.g. 'On inspection of the data we noticed a trend of increasing dive time through the encounter; a post-hoc Spearman rank correlation test was significant') - also specify either that no test was carried out for the other whale or report its results.

L229 the relation between recording time and number of spectrograms is a bit mysterious here

L231 this implies the same vocalisations but in a different order ('sequence') - is this what was meant?

L236 Table 1 I am now more convinced that 'single clicks' is a misnomer here - there are a lot of them!

L248-249 - this sounds like normal foraging echolocation - better reference to existing literature should be made here e.g. Gannier and collegues' analyses of foraging sperm whale vocalisations

L251 there appears to have been some frequency domain signal processing analysis here but the details are miissing please provide full methodological details for recording the click bandwidths - e.g. was it a -10dB standard bandwidth for transients? Or a spectral analysis, in which case to what resolution? Was the range 'eyeballed' off the spectrogram? If so were multiple observers used to check for consistency? I think we can expect that these methodological checks should be performed for publication in a journal of Plos One stature.

L257-263 agains these tests are poorly described and poorly justified - why do we care about these details? No framework of hypotheses was introduced, these just give the impression of being run ad-hoc after viewing the data, which is fine but should be labelled as such to avoid giving misleading ideas about the strength of the statistical findings.

L300 unclear how these rates were calculated - if there were other types of codas produced between examples of 'coda_1' then how is this meaningful (i.e. if the sequence was coda_1, coda_2, coda_1 why does it make sense to measure the rate of coda_1 occurence? No justification is provided, which gives the ijmpression the theoretical framework of this paper requires more work.

L302 this is confusing me between measure of coda duration and of inter-coda intervals

L324 'gain a better understanding' is quite vague - what specifically did we gain in terms of understanding? What do we understand now that we did not before?

L328 A rather obvious possibility for the different behaviours seems to have been missed here - is it not possible the less active whale had been entangled longer before encounter? Mention is made of numerous injuries and lacerations which suggests a prolonged interaction and perhaps the animal was just exhuasted?

L346 what is the difference between ''surfacing' and 'naturally resting'?

L358-360 maybe just me but I fail to see how the data reported support the assertion here that it is tail weight rather than vessel and human proximity that is causing the elevated ventilation rate...

L373 why isn't this used before! It would make much more sense to described these as 'usual' or better as 'clicks resembling regular echolocation clicks' if there is doubt, and justify earlier in the light of the slow repetition rates...

L379 more often referred to as 'slow clicks' in contemporary literature

L383 I think the authors mean that usual clicks are usually produced in prolonged bouts interspersed with buzzes rather than the clicks themselves being a long duration vocalisation, which they are not

L403-404 more is needed beyond assertion that this was not typical socialising creak

L412 I fundamentally disagree that the data here provide the basis for this assertion of 'distress creaks' and the reference to 'fear screams' is obscure indeed whie the comparison to dolphin whistles and similar duration humpback calls is very weak. This should be struck as it a conclusoin that is fundamentally unsupported by the data.

L433 I don't accept that sufficient evidence to differentiate between a 1+2+1 pattern and a 3+1 pattern has been presented. This speculation should be removed.

L437 this speculation goes way beyond the data. It should be removed. There is nothing here that can speak to clan structure.

L448 the same is true of the speculation about alarm codas here

L462-464 the use of language like this is unlikely to lead to productive and problem solving collaborations with the fishing industry on this issue. I suggest a moderation. Also the basis for asserting that the whale as cut out of a 'live' net rather than entangled in a small amount of abdanonded 'ghost' gear is not clear...

There are numerous linguistic glitches. The manuscript needs the attention of a colleague with full professional proficiency in scientific English - here I list the errors in the introduction and abstract only but they are pervasive through the entire manuscript and need careful editing: L24 'during rescue operation' L37 'efficient standardized rescuing protocol application' L35 'follow the animal physical/psychological states' L43 'shown by genetic evidences' L45 'counting less than' L61 'the morphological aspects of the bathy-morphological setting' L63 'makes no exception' L66 'and successively stranded dead' L71 'the acoustic analysis were conducted' L73 'documented other two sperm whales’ entanglement'

6. PLOS authors have the option to publish the peer review history of their article (what does this mean?). If published, this will include your full peer review and any attached files.

Reviewer #1: No

Reviewer #2: No

Reviewer #3: No

---

## [Author Response · Author response to Decision Letter 0]

14 Mar 2021

Response to Editor’s comments

AR in this file is Authors Reply

Editor Comments:

Your manuscript received mixed reviews, and clearly a substantial edit and improvement is needed. Please revise addressing all of the comments of the reviewers, paying particular attention to reduce or more fully justify speculative comments in the discussion as pointed out by reviewer 3.

Journal Requirements:

1. Please ensure that your manuscript meets PLOS ONE's style requirements, including those for file naming. The PLOS ONE style templates can be found at:

AR: Thank you for providing us the necessary information to meet journal formatting requirements. We change our format accordingly.

2. We note that Figure 1 in your submission contain map images which may be copyrighted. All PLOS content is published under the Creative Commons Attribution License (CC BY 4.0), which means that the manuscript, images, and Supporting Information files will be freely available online, and any third party is permitted to access, download, copy, distribute, and use these materials in any way, even commercially, with proper attribution. For these reasons, we cannot publish previously copyrighted maps or satellite images created using proprietary data, such as Google software (Google Maps, Street View, and Earth). For more information, see our copyright guidelines: http://journals.plos.org/plosone/s/licenses-and-copyright.

2.1. You may seek permission from the original copyright holder of Figure 1 to publish the content specifically under the CC BY 4.0 license.

2.2. If you are unable to obtain permission from the original copyright holder to publish these figures under the CC BY 4.0 license or if the copyright holder’s requirements are incompatible with the CC BY 4.0 license, please either i) remove the figure or ii) supply a replacement figure that complies with the CC BY 4.0 license. Please check copyright information on all replacement figures and update the figure caption with source information. If applicable, please specify in the figure caption text when a figure is similar but not identical to the original image and is therefore for illustrative purposes only.

AR: thank you for your suggestions. We have replaced Figure 1 with one processed ex novo. The actual figure shows only data reproduced by ourselves on the basis of public domain layers (source: https://glovis.usgs.gov/ with personal account and https://www.gebco.net/data_and_products/gridded_bathymetry_data/#a1-) and data (source: http://www.naturalearthdata.com/).

Response to Reviewers' comments

Reviewer's Responses to Questions:

1. Is the manuscript technically sound, and do the data support the conclusions?

Reviewer #1: Yes

Reviewer #2: No

Reviewer #3: Partly

2. Has the statistical analysis been performed appropriately and rigorously? 

Reviewer #1: Yes

Reviewer #2: No

Reviewer #3: I Don't Know

3. Have the authors made all data underlying the findings in their manuscript fully available?

Reviewer #1: Yes

Reviewer #2: Yes

Reviewer #3: No

4. Is the manuscript presented in an intelligible fashion and written in standard English?

Reviewer #1: Yes

Reviewer #2: Yes

Reviewer #3: No

5. Review Comments to the Author

REVIEWER #1: REVIEW PLOSONE D-20-40631

This paper presents an interesting description of two net entangling events that occurred to sperm whales around the Aeolian Islands in the lower Tyrrhenian Sea. The authors describe the disentangling operations and the associated behaviour of the two animals during the rescue operation. The paper is pretty descriptive, and scientific results are limited to a description of occurred behaviours, rather than a scientific interpretation; however, the paper depicts the problem of illegal driftnets and gives cues and recommendations about how to face such entangling events and related rescue operations. Considering this, I recommend the publication with a minor revision to correct some inconsistencies and ambiguities in the acoustic behaviour description. Overall, it is a very interesting lecture on an unusual and poorly known event.

AR: thank you for the useful comments and revisions. This paper aims to highlight the lack of a rescuing protocol in our area and, at the same time, providing useful data on distressed sperm whales.

124: please add more information on recording: depth of the hydrophone? sampling rate of the recorder?

AR: done. We added depth, sampling rate, and sensibility of the hydrophone.

127: measure rather than “extrapolate”

AR: changed

127: frequency parameters rather than just “frequency” if you mean measures of the frequency range of the vocalizations; otherwise, if you mean “frequency of occurrence” please explain better.

AR: thank you for the explanation. We were referring to upper and lower frequency limits

243: please briefly explain the structure/pattern of the codas with references to available literature for Med codas (to be discussed in later in the paper)

AR: added coda structures and references.

247: (12 spectrograms)? what do you mean?

AR: it is actually the number of sound files that we analysed, but we realize that it is redundant information, removed

251: the description of frequency limits is not clear. Use the expression “lower frequency limit” and “upper frequency limit”. Or just refer to the "frequency range". Also explain how the frequency range has been calculated, e.g. at -XX dB ref max amplitude in the spectrum?

AR: we reported the range of the lower and the upper limits of the frequencies detected. We did not calculate the frequency range.

253: “click frequency rate” is incorrect: to avoid confusion don’t use the term “frequency” when you write about “rate”

AR: removed “frequency”

256: same as 251

AR: thank you, we corrected here as well

258: “frequency” do you mean “frequency range”?

AR: we meant the mean frequency as result of the upper and the lower limit measured in Raven, but we have few data to do a comparison so we removed from the revised manuscript this result.

264, 277, 279: it is not clear the meaning of the number in parentheses

AR: it is the SD journal format. For more clarity, we added (SD), before the actual figures

301: per second … for second … please correct

AR: corrected

Discussion

interesting to note that your codas are different from the classical 3+1 coda reported in literature as the most common in the Med. However the 2+1 has been already reported in the Med. Also interesting to note the production of different codas by the same individual.

AR: Thank you for your interest to these data. We found very interesting as well the production of different codas by the same individual, and we are going to analyse coda types from different individuals in the same area.

428: Pace et al. in parentheses rather than referenced by a number.

AR: corrected

Biblio:

18. For the Italian strandings database it is also worth to mention this paper, unfortunately in Italian, but the only available to describe the data bank:

Pavan G., Bernuzzi E., Cozzi B., Podesta M., 2013. La rete nazionale di monitoraggio degli spiaggiamenti di mammiferi marini. Atti 44 Congresso SIBM. Biol. Mar. Mediterr. (2013), 20: 262-263.

AR: substitute to the old reference

REVIEWER #2: GENERAL COMMENT:

The authors present two rescue operations of sperm whales entangled in driftnets in the Aeolian Archipelago. Behavioral data of both sperm whales were collected over the rescue operations. This study is not a research paper where you expect a clear biological question and associated reproductible experimental design with sufficient sample size to answer your hypothesis. This work corresponds to a case report paper which provides informative and useful material for future similar rescue cases and for supporting regulations related to illegal use of driftnets at risk for whales’ entanglements. I recognize the great effort that was accomplished here with both the rescue process and the collection of behavioral data. This represents an obviously important piece of information, from a conservation point of view, and also for enriching our knowledge on behavioral features displayed by sperm whales under particularly stressful contexts. The manuscript contains data that deserve to be spread out into the scientific community, however, it does meet the criteria for a publication in PLOS ONE (2 report cases, and see specific comments). May I suggest a more appropriate publication of this work as a report paper in "Journal of Cetacean Research and Management" or as in "Aquatic Mammals". I list below some specific comments and suggestions that I hope will be useful to improve some aspects of the manuscript.

AR: thank you for the comments and suggestions. We appreciate your advice, even though we believe that PLOS ONE is a more suitable journal where to publish our work. This paper is not only a case study paper including the behaviour of these two entangled animals, but it reports data that can be used to properly address rescue operation outputs or as a reference point for other studies. The cited journals are too specific in our opinion and the take-home message of our paper could be lost.

Specific comments:

1) Objectives and focus of the paper

In the introduction, the authors provide a rich documentation on the issue of illegal use of driftnet, highlighting well the conservation problem for the species. However, a followed clear objective is missing.

I found attempts to state objectives/hypothesis to be tested here:

- in the introduction, L. 79-83 (“The study of the phases following the rescue operations can provide … increasing the chances of post-release survival.”)

- in the discussion, L 324-327 (“Combining behavioural, ... to reduce additional stress or accidental injuries during rescue operations.”).

AR: We have expanded the objectives and focus the paper as suggested.

L. 482. What would you do during a rescue operation if acoustical features associated to an increased-stress would occur? I don’t think that such monitoring sound production would make a rescue team to decide to interrupt the rescue operation.

AR: we appreciate your comment, but we believe that acoustics could be useful to warn the operators to use more caution or decrease the number of people around the animal or understand if the animal is getting nervous or not. We state it in the recommendation section.

There is not enough material to accurately test whether acoustics can allow monitoring stress level, neither to estimate whether the protocol allowed for more or less chances of post-release survival. I recommend to maintain the focus on the actual key outcomes of the paper: i) the rich description of behavioral displays (including acoustic data) exhibited by sperm whales in particularly stressful conditions (L. 77-78, L. 321-322), and ii) the list of valuable recommendations based on this experience study, that should be useful for potential future rescue operations (§L. 470).

AR: thank you for the feedback. We removed from the revised manuscript any assumption that is not sustained by data.

The descriptive analysis of the behavioral change over the rescue phases is interesting to document (L. 208 and L. 253 « In particular, a higher breath rate was recorded during ongoing rescue operations for both whales”, “…the click rate was higher during ongoing rescue operations”). However, to my opinion, they do not represent the most interesting findings; the paper would have overall more value and would be also attractive for a broader reader community by shaping/accentuating the focus into: providing new data on the behavioral pattern (especially on acoustic behavior) displayed by sperm whales under particularly stressful conditions (entangled, rescue operation) and comparing it to the existing literature, i.e. ref. on the typical natural behavior (e.g. how is it common to produce creaks for juvenile sperm whales there and for the species in general…etc), and ref. on the behavioral pattern exhibited in other stressful contexts such as in response to an increased predation risk or other types of disturbances (e.g. man-made noise) for which there is an existing literature.

AR: thank you for the advice, we added this part at the beginning of the discussion session along with references.

2) I don’t see the relevance to statistically compare the behavior of both whales (e.g. L. 282: comparison of number of creaks between both whales): the sample size is critically low (n=2), there is no data on the whales’ baseline behavior to assess for a natural inter-individual variability, the timing at which whales had been entangled when discovered is unknown, and there were additional factors (e.g. inconsistencies of the rescue operation phases across whales, external factors like number of boats present in the area etc) that might also have influenced the whales’ behavior. Given this highly variable context and factors, I would not focus on comparing both whales in the results section but rather would keep this in a descriptive way for the discussion, with proposing some explanation for such differences (as done in L. 366).

AR: we appreciate your suggestion and we agree with your comment including in the new version of the manuscript the possibility that different factors may be involved in the observed behavioural pattern. As suggested, we removed the behavioural comparison between whales in the results section. However, we retain the acoustical comparison, because we believe that it could be useful for future studies and underline eventual inter-individual differences.

3) comparison across the rescue phases.

AR: We confirmed our results using a KW followed by a pairwise Mann-Witney test with Bonferroni correction.

If the authors aim to make comparison (even a descriptive one) between whales through the different phases, it would help to provide more details of each phase and clarify the level of consistency of the phases between the whales.

-e.g. L. 103, and Figure 4: “monitoring” phase for whale #2 is actually a post-release phase with presence of operators tracking the whale in order to attempt to remove a remaining piece of net. I suggest to delete the term “monitoring” phase and instead have the “post-release” phase with (for whale #2) or without (for whale #1) tracking.

-L. 100: it is only said here that during pre-release period the “operators approached the whale” but there was also a rope that was attached to the tail as mentioned later in the text (Fig 2, and L. 144-146).

AR: We rewrote that part in order to clearly state the consistency of the phases between whales. We accept your suggestion to rename some phases, agreeing that “monitoring” is not accurate. We change that in “Following”, because in that phase we actually followed the escaping whale. We also change “ongoing” in “rescuing”, “pre-release” in “pre-rescue” and “post-release” in “post-rescue”, more in line with the description of the main event happened in that phase.

4) interpretation.

Figure 4: Do you have reference to support that those behavioral displays are stress-related? For the increased breath rate, your justification supporting the stress state is more furnished and thus more convincing (L345-364) that for the other behavioral parameters (acoustics, and surface displays).

AR: we removed any reference to stress-related. We agree that although our data seems to support this hypothesis, the sample size is too limited. 

L. 27: “ …it collaborated during rescue operations showing only few behavioural evidences of stressful condition.”

Getting nearly quiet and hardly moving could actually indicate an extreme degree of stress. So I would not restrict the “stress-related” indicators to the agitation and movements displayed by the Whale number 2.

AR: we removed that sentence.

There are a considerable list of references but the findings are not well shown in the light of them (e.g. with what we know on the acoustic behaviour of sperm whales). I suggest to clearly present what is known on the species regarding the typical natural breath rate, and the known stressful behavioral contexts for which particular displays such as codas were shown to be produced (e.g. anti-predatory responses, etc.); this would consolidate the interpretation of the behavioral data analysis.

AR: As suggested in the revised manuscript we better clarified the typical natural breath of a sperm whale during surfacing and before diving in the discussion. The behavioural contexts in which codas are produced is in the new version of the manuscript clearer referring to a lot of bibliography. In addition, as our codas were herd from whales in a stressful behavioural context, we preferred to keep the discussion in terms of the possible role of some behavioural/acoustic factors to non-invasively measure the stress levels of entangled whales.

L. 336-337- (ref #58): this is a ref of dolphins in zoo-captivity that showed agonistic interactions with conspecifics. Could you detail how it supports (e.g. common aspects) what was observed in the sperm whales’ behavioral displays here?

AR: We retained the reference. The reference reports an example of agonistic behaviour especially used for threat conspecific. We use this reference to suggest another interpretation of the behavioural displays observed in the study that should be threatening the rescuers.

- L. 424. and L. 449. In addition to ref 41, it could be mentioned that codas can be produced in a context of anti-predator behavior which represent a particularly stressful context for the whales (e.g. in Curé et al. 2013. Responses of male sperm whales to killer whale sounds: implications for anti-predator strategies. Sci Rep. 3:1579. doi: 10.1038/srep01579.)

AR: thank you for the suggestion. Added (ref #101)

- See also this additional ref for the discussion part on clicks: Tønnesen et al. 2020. The long-range echo scene of the sperm whale biosonar. Biol Lett. 16(8):20200134. doi: 10.1098/rsbl.2020.0134.

AR: we appreciate the advice. We added this reference as a substitute for ref#79, which was quite outdated (1966). 

Other comments:

- L. 75, 78: “Acoustic vocalizations”. Vocalizations are “acoustics” so you can delete it. I’d actually rather talk about “acoustic behavior of” or “sounds produced”.

AR: done in the whole manuscript.

- L. 87 and L. 93: the authors mention the first whale as a “10m juvenile male sperm whale”. Does the gender and estimated age class can be also mentioned for the second 9m whale? Also, a more furnished description of the physical aspects of the two animals (body condition) would feed further potential explanations for the different behaviors between whales.

AR: We added the requested information.

- Figure 4: Why to separate the pre-release and ongoing phases if data are pooled together at the end? 

AR: Because the pre-release phase includes rescue preparations (i.e. secure the animal tail with a rope), which has not happened for the second whale. In the revised version of the manuscript (methods) a new classification of the different phases was done including those that were in common for both whales and those that were only for one.

- L. 367: Data do not suggest that. These are only hypotheses proposed to explain why differences are observed.

AR: removed

Reviewer #3: Behaviour and acoustic vocalizations of two sperm whales (Physeter macrocephalus) entangled in illegal driftnets in the Mediterranean Sea Review of PONE-D-20-40631

This paper reports some behavioural observation of two sperm whales during operations to attempt to disentangle them from driftnet fishing gear. The observations are interesting and the lessons learned could potentially be applied to other similar situations. Unfortunately, the manuscript as presented is not suitable for publication. The main problems are (1) a lack of clarity and precision in the descriptions of data collection, processing, and analysis, especially a lack of clarity over which hypothesis tests were specified a priori and why and which were conducted post-hoc having viewed the data and thus carrying much less weight. The distinction must be made clear in any revision. (2) A series of speculative and unsupported claims about vocalisation function in the discussion which are really not supported at all by the data collected, which of its nature simply is not fit to test these kinds of ideas being entirely opportunistic and observational and (3) the need to improve the written presentation with the help of a colleague with full professional proficiency in scientific English.

AR: thank you for the feedback, comments and suggestions. We agree with most of your comments and we strongly revised the manuscript to make it more suitable for publication. (1) We agree that some sections are not clearly explained and some terms are used incorrectly, making the reading flow very confusing and changed them in the revised version. (2) We agree with that too and removed speculations and unsupported hypothesis from the discussion (3) The scientific English of the manuscript was revised by a full-proficiency colleague before resubmission.

An ethics statement should be provided. No justification is provided for restrictions to data availability - this does not seem commensurate with contemporary scientific expectations - I do not accept that 'all relevant data are within the manuscript'. In my eyes, this compromises the integrity of the work.

AR: We claim in the acknowledgments that any ethics statement needs to be provided because we do not handle any animal. However, we state the owned permits in the acknowledgments too. About data restriction, we disagree. Our datasets derive from two emergency events, so may be limited by rescue operations and the difficulty of the situation (moving the boat, collaborate with the rescue team, etc.). We confirm that all relevant data recorded in this study are included in this manuscript.

L19 under which jurisdiction? Italy? EU?

AR: Rewrote: in different Mediterranean areas, primarily in the Tyrrhenian Sea

L23 it is an opportunity to study 'whales in extremely stressful conditions' but why more broadly is that important?

AR: thank you for the comment. We rewrote the sentence.

L31 is 'single clicks' what are called regular clicks in the rest of the literature? Better to keep consistent

AR: thank you for the suggestion, but we prefer to retain ‘single click’ in the revised manuscript due to the particular nature of this study. We cannot assume a priori that those sounds could be what literature call ‘regular clicks’ or “slow clicks” for the characteristics of this vocalizations. Particularly, in this study, the term ‘single clicks’ is used to indicate those sounds which resembling regular echolocation clicks (“usual clicks”) associated primarily with a foraging behaviour but with some substantial differences with surface clicks or slow clicks (unknown behavioural function and peculiar ICI). We better clarify clicks classification in the revised manuscript and compared results with the available literature. 

L33 'associating' is vague

AR: changed in ‘comparing’.

L35 I do not support the assertion of 'alarm' and 'distress' in these vocalisations - describe the form but extra evidence is needed to ascribe function

AR: We remove that part and other speculations from the revised manuscript.

L47 there are more recent IUCN assessments of this threat - it is changing and better to reflect the most recent assessments see Notarbartolo et alarm

AR: thank for the correction. Changed reference #3 as suggested.

L55 'national database' - which nation!? Are stranding records from 1714 really thought comparable to those in the last two decades? Maybe revise this to a more pertinent timescale.

AR: changed ‘national’ with ‘Italian’ and reduced the timescale to the last two decades (2000-2020).

L57 why is the location relevant? isn't it the cause we are more concerned about?

AR: In the area is not already instituted a Marine Protected Area so we want to give more importance to the location context and critical situation.

L69 why are vocalisations not a behavioural observation? Also 'acoustic vocalisation' is redundant - there aren't any non-acoustic vocalization, right?

AR: removed or changed with ‘acoustic behaviour’.

L76 'immediately after release, and in the hours-days thereafter' the distinction between these two categories is not clear

AR: removed, it is clearly explained in M&M.

L77-78 see point above from abstract - yes a great opportunity but to what wider relevance?

AR: we rewrote this section and explained better the aim of the study.

L80 '(i.e. acoustic parameters' this is not a method

AR: changed ‘parameters’ into “recordings”.

L82-83 this level of effort is surely only justified if the fate of one or two individuals over a 5-10 years period will make a population level difference - I don't think Med sperm whales are that badly off yet. Perhaps the argument for preparedness could be used - not a major problem yet but could become vital if population trajectories continue downward

AR: removed ‘thus increasing the chances of post-release survival‘

L86 I think 10:01 UTC+1 should be the standard time format throughout but the journal may have style rules

AR: The journal does not provide any specific time format, so we are accepting your suggestion. Thank you.

L87-94 it would be interesting to know if the nets appeared superficially similar suggesting they sourced from the same fishing operation or different

AR: Unfortunately, we did not have the opportunity to do this investigation because the nets were subtracted by the Coast Guard.

L93 no sex information for the second whale?

AR: It seems to be a male but we don’t have a clear image to confirm this. We kept the distance for not disturbing the animal and the rescue operations during the two events. The second whale was very nervous and from a distance was not possible for us to identify the sex with the GoPro videos. We asked to a video-operator, one of the people not involved in rescue operations which was very close to the individual (and cause of stress for him) to provide us some screen shots of the ventral area in order to identify the sex but he did not reply to our request. The videos of this video maker were used to produce a documentary for the television but not for scientific purposes.

L96-97 it is not necessary to describe operations down to the level of plotting points on a map - if the GIS was used to produce a particular piece of data from a geographic database, fair enough.

AR: removed

L100-101 what was 'pre-release' only relevant for the first whale? What was the rationale for a distinctive timeline phase? More information is needed on why different timelines were used for each whale.

AR: We agree that this section is very confusing. We have rewritten the sentence.

L108 I think this is best written as 'counting breaths per minute' since focal observations sampling at one-minute intervals strongly suggests point sampling which I don't think it what the authors did since they refer to breathes per minute later

AR: We better explained and corrected this sentence. Both breaths and behaviour were recorded at one-minute intervals and these were sampled at the same time. We recorded the number of behavioural events for minute and for the entire duration of each phase.

L113 this is confusing because the activities listed are events - the animal performed a specific behaviour like side-roll or spy hop - so how are 'percentage of each activity' calculated specifically?

AR: The behaviour of the sperm whales was recorded at one-minute intervals (sets). We recorded the number of behavioural events for each set. The percentage of each activity was calculated as number of events/number of sets in the different phases.

L118-119 no pairwise follow up in the case of a significant KW result?

AR: We performed new tests to confirm our results: Mann-Whitney test and the Bonferroni correction for KW.

L120-122 what is the theoretical justification for expecting a monotonic if non-linear correlation between time of day and length of dive? Was this hypothesis established a priori or tested after viewing the data? The difference matters!

AR: we removed the regression analysis on these data from the results.

L123 this sentence is also true if you remove 'stressed'...

AR: done.

L125 give at least the hydrophone +/-3dB frequency response here and give more detail on what the 'wideband solid-state recorder' was - sampling rate? Bit depth? Make/model?

AR: added.

L127 I don't think 'extrapolate' is the right word here.

AR: changed in ‘measured’.

L128 'single clicks' is not something I recognise from the literature - do the authors mean 'regular clicks' or 'slow clicks' as characteristically produced by males?

AR: thank you for the suggestion, we better clarify clicks classification in the revised manuscript. We prefer to retain ‘single click’ in the revised manuscript due to the particular nature of this study. We cannot assume a priori that these clicks could be what the literature call ‘regular clicks’ or “slow clicks” (added in Methods). Particularly, in this study, the term ‘single clicks’ is used to indicate those sounds which resembling regular echolocation clicks (“usual clicks”) associated primarily with a foraging behaviour but with some substantial differences with surface clicks or slow clicks (unknown behavioural function and peculiar ICI). In the discussion we added these considerations and compare the characteristics of these vocalizations with those found in other studies. 

L169 what is a 'lighting boa'? A lit mark of some kind?

AR: changed ‘lighting boa’ in ‘lit mark buoy’.

L171 I think perhaps 'long dives' rather than 'deep' would be more precise here?

AR: changed as suggested. We don’t know how deep the whale descended.

L173-174 this is kind of vague and speculative - what is a 'great exploration effort'? Better to say 'X boats, Y planes and Z individuals

AR: we removed that sentence, because we don’t know the actuals figures of this effort.'

L198 this is exactly the same information that started the previous section - avoid this repetition

AR: we used the same method for both behavioural and respiratory patterns, but, yes, we agree that it could be confusing. Rewrote by referring to the behavioural data.

L210-211 I am unclear what is being tested here - a direct comparison is suggested by the text specifically between whale 1 vs whale 2, in which case only pairwise tests are needed? So, what was the multiple sample KW test used for? Please provide more clarity here.

AR: We better explained in methods and results

L220 pretty sure this should be 'breath/minute' - if you count the breathes in a minute then divided by 60 then you get breath/second... please check and clarify

AR: sorry, it was a misspelled error, corrected.

L216-217 again I am unclear how these tests are working - the KW test gives you a test of whether all the samples come from the same population but doesn't specify which categories vary, for that you need post-hoc pairwise tests

AR: We confirmed our results running a KW followed by Mann-Whitney test with Bonferroni correction.

L220 this section basically confirms to me that the test of the dive time vs time of day was a post-hoc hypothesis test and should be labelled as such (e.g. 'On inspection of the data we noticed a trend of increasing dive time through the encounter; a post-hoc Spearman rank correlation test was significant') - also specify either that no test was carried out for the other whale or report its results.

AR: we rewrote this section according to our suggestion. Thank you.

L229 the relation between recording time and number of spectrograms is a bit mysterious here

AR: We apologize, spectrograms were meant as the number of files analysed. It is a redundant information, as we already have the recording time.

L231 this implies the same vocalisations but in a different order ('sequence') - is this what was meant?

AR: yes

L236 Table 1 I am now more convinced that 'single clicks' is a misnomer here - there are a lot of them!

AR: Already explained before.

L248-249 - this sounds like normal foraging echolocation - better reference to existing literature should be made here e.g. Gannier and collegues' analyses of foraging sperm whale vocalisations

AR: thank you for the advice, we substitute ref#51 with your suggestion.

L251 there appears to have been some frequency domain signal processing analysis here but the details are missing please provide full methodological details for recording the click bandwidths - e.g. was it a -10dB standard bandwidth for transients? Or a spectral analysis, in which case to what resolution? Was the range 'eyeballed' off the spectrogram? If so were multiple observers used to check for consistency? I think we can expect that these methodological checks should be performed for publication in a journal of Plos One stature.

AR: We believe that those analysis weren’t pertinent with the aim of the paper, which is a descriptive report of two entanglement events. Those kinds of data would be used for a more complete study of vocalizations of foraging sperm whales in the area.

L257-263 against these tests are poorly described and poorly justified - why do we care about these details? No framework of hypotheses was introduced, these just give the impression of being run ad-hoc after viewing the data, which is fine but should be labelled as such to avoid giving misleading ideas about the strength of the statistical findings.

AR: Better described and more justified the statistic in the revised manuscript. KW tests with correction and Mann-Whitney included in M&M and results

L300 unclear how these rates were calculated - if there were other types of codas produced between examples of 'coda_1' then how is this meaningful (i.e. if the sequence was coda_1, coda_2, coda_1 why does it make sense to measure the rate of coda_1 occurrence? No justification is provided, which gives the impression the theoretical framework of this paper requires more work.

AR: maybe this part was a bit misleading. Coda 1 was the first sequence of coda emissions between the two creak events. Coda 2 was the second sequence of coda emission after the second creak sequence. So, we calculated the coda rates for the two sequences and compare them.

L302 this is confusing me between measure of coda duration and of inter-coda intervals

AR: we apologize, we used some terms unproperly. Corrected.

L324 'gain a better understanding' is quite vague - what specifically did we gain in terms of understanding? What do we understand now that we did not before?

AR: we change ‘better’ with ‘wider’.

L328 A rather obvious possibility for the different behaviours seems to have been missed here - is it not possible the less active whale had been entangled longer before encounter? Mention is made of numerous injuries and lacerations which suggests a prolonged interaction and perhaps the animal was just exhausted?

AR: yes, we explain it afterward.

L346 what is the difference between ''surfacing' and 'naturally resting'?

AR: none. Deleted.

L358-360 maybe just me but I fail to see how the data reported support the assertion here that it is tail weight rather than vessel and human proximity that is causing the elevated ventilation rate...

AR: we agree, it was a speculation. Removed.

L373 why isn't this used before! It would make much more sense to described these as 'usual' or better as 'clicks resembling regular echolocation clicks' if there is doubt, and justify earlier in the light of the slow repetition rates...

AR: thank you for the suggestion. Added a statement in ‘single clicks’ results section. 

L379 more often referred to as 'slow clicks' in contemporary literature

AR: added specification for slow clicks.

L383 I think the authors mean that usual clicks are usually produced in prolonged bouts interspersed with buzzes rather than the clicks themselves being a long duration vocalisation, which they are not

AR: thank you for the specification. Changed.

L403-404 more is needed beyond assertion that this was not typical socialising creak

AR: we removed that sentence and rewrote the entire section, in order to be more consistent to data and literature.

L412 I fundamentally disagree that the data here provide the basis for this assertion of 'distress creaks' and the reference to 'fear screams' is obscure indeed while the comparison to dolphin whistles and similar duration humpback calls is very weak. This should be struck as it a conclusion that is fundamentally unsupported by the data.

AR: we moderate our statements to make them more consistent with data, but we are not excluded this possibility.

L433 I don't accept that sufficient evidence to differentiate between a 1+2+1 pattern and a 3+1 pattern has been presented. This speculation should be removed.

AR: thank you for the suggestion. Removed.

L437 this speculation goes way beyond the data. It should be removed. There is nothing here that can speak to clan structure.

AR: we agree. Removed

L448 the same is true of the speculation about alarm codas here

AR: as L412 comment, we moderate our statements to make them more consistent with data, but we are not excluded this possibility.

L462-464 the use of language like this is unlikely to lead to productive and problem-solving collaborations with the fishing industry on this issue. I suggest a moderation. Also the basis for asserting that the whale as cut out of a 'live' net rather than entangled in a small amount of abdanonded 'ghost' gear is not clear...

AR: thank you for the suggestion. We removed that phrase. The basis of asserting that the whale cut out a live net is solid, because that kind of net is illegal in the Mediterranean Sea since 1998. So, it can’t be abandoned.

There are numerous linguistic glitches. The manuscript needs the attention of a colleague with full professional proficiency in scientific English - here I list the errors in the introduction and abstract only but they are pervasive through the entire manuscript and need careful editing: L24 'during rescue operation' L37 'efficient standardized rescuing protocol application' L35 'follow the animal physical/psychological states' L43 'shown by genetic evidences' L45 'counting less than' L61 'the morphological aspects of the bathy-morphological setting' L63 'makes no exception' L66 'and successively stranded dead' L71 'the acoustic analysis were conducted' L73 'documented other two sperm whales’ entanglement'

AR: Thank you, we will surely submit the entire manuscript to an English full-proficiency colleague.

6. PLOS authors have the option to publish the peer review history of their article (what does this mean?). If published, this will include your full peer review and any attached files.

Do you want your identity to be public for this peer review? For information about this choice, including consent withdrawal, please see our Privacy Policy.

Reviewer #1: No

Reviewer #2: No

Reviewer #3: No

---

## [Editor Report · Decision Letter 1]

24 Mar 2021

PONE-D-20-40631R1

Behaviour and vocalizations of two sperm whales (Physeter macrocephalus) entangled in illegal driftnets in the Mediterranean Sea

PLOS ONE

Dear Dr. Blasi,

Thank you for submitting your manuscript to PLOS ONE. After careful consideration, we feel that it has merit but does not fully meet PLOS ONE’s publication criteria as it currently stands. Therefore, we invite you to submit a revised version of the manuscript that addresses the points raised during the review process.

You have addressed positively the substantial suggestions of the reviewers.  However, the article still contains English errors and one unsupported claim that should be de-emphasized in your manuscript.  I am therefore providing herein a detailed list of changes required for the manuscript to be acceptable, both in terms of content and presentation.   Hopefully all of the suggestions will be agreeable to you, but if not, please detail point by point in your response letter any that you do not accept. 

We look forward to receiving your revised manuscript.

Kind regards,

Patrick J. O. Miller

Academic Editor

PLOS ONE

Journal Requirements:

Additional Editor Comments (if provided):

You have addressed positively the substantial suggestions of the reviewers. However, I find that the article still contain errors and one unsupported claim that should be de-emphasized in your manuscript. I am therefore providing herein a detailed list of changes required for the manuscript to be acceptable, both in terms of content and presentation.

Major content change. Your observations were important, but limited. There is not sufficient scientific content in your report to ascribe reasons for outcomes in the two release attempt scenarios. While it might be that the presence of more boats was a factor causing higher responsiveness in the second whale, you cannot test that scientifically. Therefore, I suggest you cut the final sentence from the abstract, and significantly shorten the “Recommendations” section related to guidance of rescue protocols. I suggest any recommendations you state here should follow those of the Hamer and Minton publication. For example, on pg 13, those authors state “a quieter environment (less situational stress)… will increase the likelihood of survival…”.

Therefore, I suggest the recommendations section be cut down to a simple paragraph in which you emphasize guidance already published (citing the source) as much as possible. The one specific contribution your study makes is the benefit of including a hydrophone to observe the sounds of the whales during release attempts, as these might be helpful to interpret the state of the animal during operations.

The other detailed suggestions below are based upon my detailed review of the manuscript. I hope you will be able to accept them all.

Ln 26: change ‘provide’ to ‘collect’

Ln 27: change ‘notably’ to ‘likely’

Lns 31, 32, and throughout: change ‘breath rate’ to ‘breathing rate’

Ln 31 – report here the observed breathing rate for the first animal

Ln 32 – report here the observed breathing rate for the second animal

Ln 36 – after ‘single clicks’ insert “ – likely either slow clicks or regular clicks” ((this is necessary as ‘single click’ is not an accepted form of click type in the sperm whale literature, as pointed out by reviewer 3))

Lns 36-37: change “By comparing all data it was found that” to “Our observations indicate that”

Ln 37: change “animal’s physical/physiological status” to “physical/physiological status of sperm whales” -- this is the key scientific contribution of this report.

Delete the last sentence in the abstract, which is 1) controversial and 2) not scientifically supported by the paper. This sort of suggestion should only be mentioned in the discussion where some ancillary material is acceptable. I suggest you use positive wording for this point.

Ls 45 – change “Atlantic one” to “Atlantic population”

Ln 46 – change ‘this population’ to ‘the Mediterranean’

Ln 48 – insert “a” before “concerning matter”

Ln 48 – interest ‘the’ before “international”

Ln 65 – change ‘makes’ to ‘is’

Ln 74 – add ‘as sounds could not be ascribed to individual whales’ after ‘social unit’

Ln 77 – add ‘production’ after ‘acoustic’

Lns 79-80 – delete ‘intense’ before ‘stressful’

Lns 80 – 86 delete these two sentences here, as they belong in the discussion. The text to this point provides adequate motivation and background.

Ln 96 – change ‘shown’ to ‘had’

Ln 99 – change ‘Genre’ to ‘Sex’

Ln 105 – use past tense to refer to your specific study results, so change ‘are’ to ‘were’

Ln 106 – change ‘are described as follow’ to ‘were’

Ln 108 – change ‘specific phases for’ to ‘phases specific to’

Ln 115 – change ‘ has been observed, which was described as’ to ‘was observed: ’

Ln 117 – change ‘has been’ to ‘were’

Ln 123 – change ‘has been’ to ‘was’

Ln 129 – change ‘percentage’ to ‘rate’ (this is number per unit time). It should be reported as a number per unit time

Ln 136 – delete ‘mainly’

Ln 149 – delete ‘for’

Lns 149-150 change ‘for “single clicks” are intended sequences of clicks (ICI > 1 s) , not organized in a defined structure, similar to ‘usual click’ described in literature [39]’ to ‘ “single clicks” were scored for sequences of clicks (ICI > 1 s) that were not organized in a defined structure, as described for ‘usual clicks’ [39]’

Ln 154 – delete ‘post-hoc’

Ln 172 – state here the date the second whale was sighted

Ln 176 – add ‘for small cetaceans’ after ‘recommended’

Ln 192 – delete ‘of which’

Ln 193 and Figure 4 – the methods describe this as the number of events divided by the total number of sessions. Therefore, this is a rate. The Y-axes should be number of events per unit time.

Ln 202 – add “the” before “post-rescue”

Ln 205 – add ‘remained attached’ after ‘net’

Fig 5 legend or Y-axis must state what is the number – ie “breaths/minute”

Ln 219 – add ‘bin’ after ‘minute’.

Your statistic treats each one minute bin as an independent data sample, leading to a high sample size of measurements. Therefore, you need to be clear that you are comparing breaths per BIN specifically.

Ln 238 – change ‘are’ to ‘were’

Ln 241 – Delete “The first part of dataset included 56 minutes of acoustic recordings that were analysed for” and add ‘were analysed’ after “(18 July)”

Ln 244 change ‘differed in’ to ‘differed between’

Ln 254 – delete ‘Whereas, ‘

Ln 259 – change “The second part of dataset included 1519 minutes (from 19:04 to 06:15) of acoustic recordings of second whale’ re-sighting on 5th October.” To “An additional 1519 minutes (from 19:04 to 06:15) of acoustic recordings were recorded during the second whale’s re-sighting on 5th October.”

Ln 272 – state the sample size for this KW test.

Ln 274 – change ‘first one’ to ‘first whale’

Ln 282 – change ‘has been’ to ‘were’

Ln 293 – change ‘higher trains duration’ to ‘higher train duration’

Lns 294-297. Specify the samples sizes used for all statistical tests.

Ln 303 – change ‘observed’ to ‘noted’

Ln 316 – change ‘smaller’ to ‘lower’

Fig 9 legend – Change “Coda types spectrograms” to “Spectrograms of coda types”

Ln 326 – change “3-clicks” to “3-click”

Ln 331 – change ‘a stressed’ to ‘active’

Ln 336 – change “pattern, especially on acoustic’ to “patterns, including acoustic”

Ln 338 – change “outputs” to “resulting observations”

Ln 345 – change “brought to free” to “successfully freed”

Ln 347 – change ‘run away’ to ‘swim away’

Ln 348 change “whales’s” to “whales’”

Ln 363 – add “The” before ‘First whale”

Ln 364 – add “The” before “second whale”, and add “the” before “first” on the same line

Ln 365 – change “to the hypothesis” to “with the hypothesis”

Ln 386 – change “resembling” to “resembled”

Ln 387 – change “a foraging behaviour” to “echolocation-based foraging”

Ln 389 – change ‘are very low’ to ‘were very low’

Ln 394 – change “are used” to “were used”

Ln 395 – delete “eventual”

Ln 396 – change “do not fit” to “did not fit”

Ln 399 – change ‘in accordance with the stress level’ to ‘in accordance with the activity level’

Ln 401 -change “triggered by high-stress” to “during high activity”

Ln 409 – change “sight” to “encounter” both times.

Ln 414 – use past tense. Change “are” to “were” and “share” to “shared”

Ln 415 – change “are” to “were”

Ln 416 – change “stressful” to “entanglement”

Ln 418 – change ‘are definitely’ to ‘were definitely’

Ln 421 – change ‘show’ to ‘had’

Ln 423 – add “were” before “operating”

Ln 425 – change “to high” to “with high”

Ln 434 – change ‘enough’ to ‘sufficient’ and change “meaning” to “function”

Ln 437 – change “have known” to “are known”

Ln 449 – change “are” to “were”

Ln 453 – change “are more” to “were more”

Ln 460 – change “Orchinus” to “Orcinus”

Ln 461 – change “heard” to “produced”

Ln 474 – I don’t think this is the first, nor is it important if it is the first or not. See Esch et al 2009 Whistles as Potential Indicators of Stress in Bottlenose Dolphins. Journal of Mammalogy.

Ln 484 – delete “several” before “boats”

Ln 508 – change to “Acknowledgements”

Ln 518 – change “any ethics statement is” to “no ethics statement is required”
---

## [Author Response · Author response to Decision Letter 1]

6 Apr 2021

AR in this file is Authors Reply

Response to Editor’s comments

Journal Requirements:

AR: we rechecked all the reference list and corrected eventual formatting errors, in accordance with PLOSONE guidelines.

Additional Editor Comments (if provided):

You have addressed positively the substantial suggestions of the reviewers. However, I find that the article still contains errors and one unsupported claim that should be de-emphasized in your manuscript. I am therefore providing herein a detailed list of changes required for the manuscript to be acceptable, both in terms of content and presentation.

Major content change. Your observations were important, but limited. There is not sufficient scientific content in your report to ascribe reasons for outcomes in the two release attempt scenarios. While it might be that the presence of more boats was a factor causing higher responsiveness in the second whale, you cannot test that scientifically. Therefore, I suggest you cut the final sentence from the abstract, and significantly shorten the “Recommendations” section related to guidance of rescue protocols. I suggest any recommendations you state here should follow those of the Hamer and Minton publication. For example, on pg 13, those authors state “a quieter environment (less situational stress)… will increase the likelihood of survival…”.

Therefore, I suggest the recommendations section be cut down to a simple paragraph in which you emphasize guidance already published (citing the source) as much as possible. The one specific contribution your study makes is the benefit of including a hydrophone to observe the sounds of the whales during release attempts, as these might be helpful to interpret the state of the animal during operations.

AR:

The other detailed suggestions below are based upon my detailed review of the manuscript. I hope you will be able to accept them all.

AR: thank you for your suggestions, we agreed with them all.

Ln 26: change ‘provide’ to ‘collect’

AR: changed

Ln 27: change ‘notably’ to ‘likely’

AR: accepted

Lns 31, 32, and throughout: change ‘breath rate’ to ‘breathing rate’

AR: accepted

Ln 31 – report here the observed breathing rate for the first animal

AR: added

Ln 32 – report here the observed breathing rate for the second animal

AR: added

Ln 36 – after ‘single clicks’ insert “ – likely either slow clicks or regular clicks” ((this is necessary as ‘single click’ is not an accepted form of click type in the sperm whale literature, as pointed out by reviewer 3))

AR: added

Lns 36-37: change “By comparing all data it was found that” to “Our observations indicate that”

AR: changed

Ln 37: change “animal’s physical/physiological status” to “physical/physiological status of sperm whales” -- this is the key scientific contribution of this report.

AR: changed

Delete the last sentence in the abstract, which is 1) controversial and 2) not scientifically supported by the paper. This sort of suggestion should only be mentioned in the discussion where some ancillary material is acceptable. I suggest you use positive wording for this point.

AR: deleted

Ls 45 – change “Atlantic one” to “Atlantic population”

AR: changed

Ln 46 – change ‘this population’ to ‘the Mediterranean’

AR: changed

Ln 48 – insert “a” before “concerning matter”

AR: added

Ln 48 – interest ‘the’ before “international”

AR: added

Ln 65 – change ‘makes’ to ‘is’

AR: changed

Ln 74 – add ‘as sounds could not be ascribed to individual whales’ after ‘social unit’

AR: added

Ln 77 – add ‘production’ after ‘acoustic’

AR: added

Lns 79-80 – delete ‘intense’ before ‘stressful’

AR: added

Lns 80 – 86 delete these two sentences here, as they belong in the discussion. The text to this point provides adequate motivation and background.

AR: deleted

Ln 96 – change ‘shown’ to ‘had’

AR: changed

Ln 99 – change ‘Genre’ to ‘Sex’

AR: changed

Ln 105 – use past tense to refer to your specific study results, so change ‘are’ to ‘were’

AR: changed

Ln 106 – change ‘are described as follow’ to ‘were’

AR: changed

Ln 108 – change ‘specific phases for’ to ‘phases specific to’

AR: changed

Ln 115 – change ‘has been observed, which was described as’ to ‘was observed:’

AR: changed

Ln 117 – change ‘has been’ to ‘were’

AR: changed

Ln 123 – change ‘has been’ to ‘was’

AR: changed

Ln 129 – change ‘percentage’ to ‘rate’ (this is number per unit time). It should be reported as a number per unit time

AR: changed

Ln 136 – delete ‘mainly’

AR: done

Ln 149 – delete ‘for’

AR: done

Lns 149-150 change ‘for “single clicks” are intended sequences of clicks (ICI > 1 s), not organized in a defined structure, similar to ‘usual click’ described in literature [39]’ to “single clicks” were scored for sequences of clicks (ICI > 1 s) that were not organized in a defined structure, as described for ‘usual clicks’ [39]’

AR: changed

Ln 154 – delete ‘post-hoc’

AR: done

Ln 172 – state here the date the second whale was sighted

AR: done

Ln 176 – add ‘for small cetaceans’ after ‘recommended’

AR: added

Ln 192 – delete ‘of which’

AR: deleted

Ln 193 and Figure 4 – the methods describe this as the number of events divided by the total number of sessions. Therefore, this is a rate. The Y-axes should be number of events per unit time.

AR: changed

Ln 202 – add “the” before “post-rescue”

AR: added

Ln 205 – add ‘remained attached’ after ‘net’

AR: added

Fig 5 legend or Y-axis must state what is the number – ie “breaths/minute”

AR: added in the legend

Ln 219 – add ‘bin’ after ‘minute’.

AR: added

Your statistic treats each one minute bin as an independent data sample, leading to a high sample size of measurements. Therefore, you need to be clear that you are comparing breaths per BIN specifically.

AR: thank you for the suggestion, we added ‘bin’ after ‘minute’

Ln 238 – change ‘are’ to ‘were’

AR: changed

Ln 241 – Delete “The first part of dataset included 56 minutes of acoustic recordings that were analysed for” and add ‘were analysed’ after “(18 July)”

AR: delete the first phrase and added the second

Ln 244 change ‘differed in’ to ‘differed between’

AR: changed

Ln 254 – delete ‘Whereas, ‘

AR: done

Ln 259 – change “The second part of dataset included 1519 minutes (from 19:04 to 06:15) of acoustic recordings of second whale’ re-sighting on 5th October.” To “An additional 1519 minutes (from 19:04 to 06:15) of acoustic recordings were recorded during the second whale’s re-sighting on 5th October.”

AR: changed

Ln 272 – state the sample size for this KW test.

AR: added

Ln 274 – change ‘first one’ to ‘first whale’

AR: changed

Ln 282 – change ‘has been’ to ‘were’

AR: changed

Ln 293 – change ‘higher trains duration’ to ‘higher train duration’

AR: changed

Lns 294-297. Specify the samples sizes used for all statistical tests.

AR: Done

Ln 303 – change ‘observed’ to ‘noted’

AR: changed

Ln 316 – change ‘smaller’ to ‘lower’

AR: changed

Fig 9 legend – Change “Coda types spectrograms” to “Spectrograms of coda types”

AR: changed

Ln 326 – change “3-clicks” to “3-click”

AR: changed

Ln 331 – change ‘a stressed’ to ‘active’

AR: changed

Ln 336 – change “pattern, especially on acoustic’ to “patterns, including acoustic”

AR: changed

Ln 338 – change “outputs” to “resulting observations”

AR: changed

Ln 345 – change “brought to free” to “successfully freed”

AR: changed

Ln 347 – change ‘run away’ to ‘swim away’

AR: changed

Ln 348 change “whales’s” to “whales’”

AR: changed

Ln 363 – add “The” before ‘First whale”

AR: added

Ln 364 – add “The” before “second whale”, and add “the” before “first” on the same line

AR: added

Ln 365 – change “to the hypothesis” to “with the hypothesis”

AR: changed

Ln 386 – change “resembling” to “resembled”

AR: changed

Ln 387 – change “a foraging behaviour” to “echolocation-based foraging”

AR: changed

Ln 389 – change ‘are very low’ to ‘were very low’

AR: changed

Ln 394 – change “are used” to “were used”

AR: changed

Ln 395 – delete “eventual”

AR: deleted

Ln 396 – change “do not fit” to “did not fit”

AR: changed

Ln 399 – change ‘in accordance with the stress level’ to ‘in accordance with the activity level’

AR: changed

Ln 401 -change “triggered by high-stress” to “during high activity”

AR: changed

Ln 409 – change “sight” to “encounter” both times.

AR: changed both

Ln 414 – use past tense. Change “are” to “were” and “share” to “shared”

AR: changed

Ln 415 – change “are” to “were”

AR: changed

Ln 416 – change “stressful” to “entanglement”

AR: changed

Ln 418 – change ‘are definitely’ to ‘were definitely’

AR: changed

Ln 421 – change ‘show’ to ‘had’

AR: changed

Ln 423 – add “were” before “operating”

AR: added

Ln 425 – change “to high” to “with high”

AR: changed

Ln 434 – change ‘enough’ to ‘sufficient’ and change “meaning” to “function”

AR: changed

Ln 437 – change “have known” to “are known”

AR: changed

Ln 449 – change “are” to “were”

AR: changed

Ln 453 – change “are more” to “were more”

AR: changed

Ln 460 – change “Orchinus” to “Orcinus”

AR: changed

Ln 461 – change “heard” to “produced”

AR: changed

Ln 474 – I don’t think this is the first, nor is it important if it is the first or not. See Esch et al 2009 Whistles as Potential Indicators of Stress in Bottlenose Dolphins. Journal of Mammalogy.

AR: We agreed with your correction and we changed ‘is a first attempt’ with ‘is proposing’.

Ln 484 – delete “several” before “boats”

AR: deleted

Ln 508 – change to “Acknowledgements”

AR: changed

Ln 518 – change “any ethics statement is” to “no ethics statement is required”

AR: changed

---

## [Editor Report · Decision Letter 2]

16 Apr 2021

Behaviour and vocalizations of two sperm whales (Physeter macrocephalus) entangled in illegal driftnets in the Mediterranean Sea

PONE-D-20-40631R2

Dear Dr. Blasi,

We’re pleased to inform you that your manuscript has been judged scientifically suitable for publication and will be formally accepted for publication once it meets all outstanding technical requirements.

Kind regards,

Patrick J. O. Miller

Academic Editor

PLOS ONE

---

## [Editor Report · Acceptance letter]

20 Apr 2021

PONE-D-20-40631R2 

Behaviour and vocalizations of two sperm whales (*Physeter macrocephalus*) entangled in illegal driftnets in the Mediterranean Sea 

Dear Dr. Blasi:

I'm pleased to inform you that your manuscript has been deemed suitable for publication in PLOS ONE. Congratulations! Your manuscript is now with our production department. 

Kind regards, 

on behalf of

Dr. Patrick J. O. Miller 

Academic Editor

PLOS ONE